# Training a Vision-Language Model for Diverse Exploration in Open GUI World

## Abstract

Vision-language models have emerged as capable computer-use agents, showing increasing potential to automate a wide range of computer tasks through graphical user interfaces. However, their effectiveness remains bounded by a fundamental limitation: current LLM- or VLM-based agents struggle to generalize to unfamiliar applications and remain heavily dependent on large-scale, human-curated datasets. To address this, we introduce ScreenExplorer, a novel VLM-based agent designed for autonomous exploration in real, dynamic, open-ended GUI environments. Through end-to-end training with an exploration-driven objective, our approach enables sustained interaction and diverse discovery without relying on predefined task structures. Specifically, we introduce a world model-inspired curiosity reward that helps the agent to overcome the cold-start phase of exploration, coupled with state-change-based exploration rewards to encourage agent's intrinsic motivation for venturing into novel states. Additionally, an experience stream distillation mechanism is designed to systematically accumulate and refine exploratory policies, enabling continual learning from gathered experiences. Extensive evaluations demonstrate that ScreenExplorer achieves remarkable generalization and diverse exploration capabilities in unseen applications, significantly outperforming static deployment baselines. This work establishes a new paradigm for GUI agents to progressively learn through autonomous exploration, moving beyond static dataset dependency toward adaptive, lifelong learning in complex digital worlds.

## 1 Introduction

Vision-language models (VLMs) have demonstrated considerable proficiency in understanding and reasoning about visual and textual information, establishing them as a promising foundation for autonomous computer-use agents (CUAs) capable of interacting with graphical user interfaces (GUIs) Hu et al.. A central goal of computer-use agents is to operate in open-world environments where interface content and task structures are dynamic and unpredictable. To this end, enabling agents to generalize robustly across diverse and evolving open-world contexts represents a compelling yet under-explored research direction Zhang et al. (2025a); Tan et al. (2024b).

However, open-world GUI environments pose two major challenges: First, the content of the interface is dynamic. For example, pages in news websites or file browsers often change over time, requiring agents to adapt to evolving states. Second, the state space is virtually unbounded, making it infeasible to manually collect sufficient diverse data to support robust generalization. Current approaches, however, are predominantly based on frozen models that cannot update their parameters through interaction. This architectural constraint prevents them from learning from trial and error Zhang et al. (2024; 2025b); Wu et al. (2024c); Agashe et al. (2025a;b); Hong et al. (2024) and ties their performance permanently to the limited scope and high cost of human-annotated datasets Deka et al. (2017); Zhang et al. (2021b); Deng et al. (2023); Wu et al. (2024b); Niu et al. (2024); Chen et al. (2024). Therefore, the ability to effectively interact with new environments and autonomously perform diverse/ active exploration emerges as the central challenge for the evolution of GUI agents.

To address these limitations, we introduce ScreenExplorer, a VLM-based agent trained through reinforcement learning (RL) in a real, dynamic GUI environment. During training, we design state-change-based rewards to encourage actions that lead to successful interaction in novel environments.

To further incentivize exploration, we introduce a World Model(WM) that learns the environment's transition dynamics. The discrepancy between the predicted and actual state transitions serves as an intrinsic curiosity signal, measuring state novelty and guiding the agent toward under-explored regions of the state space. Additionally, we collect the experience streams generated during exploration and distill the original model. This learning strategy is designed to mitigate exploration bottlenecks and enable sustained capability improvement, aiming to lead to better initialization. As shown in Figure 1, ScreenExplorer demonstrates significant improvement from the worst to the best in exploring diversity through RL training.

Our results suggest that training agents via RL in open GUI environments fosters both effective interaction and diverse/autonomous exploration. By combining these abilities with experience stream distillation, agents can gradually reduce reliance on manually collected data and continuously evolve. Our contributions are as follows:

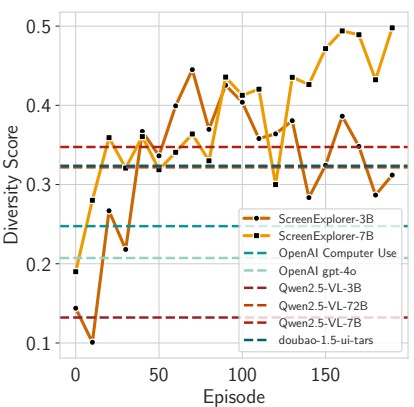

Figure 1: ScreenExplorer RL training leads to better GUI exploration diversity versus static models.

- We propose ScreenExplorer, a vision-language model (VLM) agent trained via world-model-based RL in a real, dynamic, and open-ended GUI environment. The agent is rewarded for both successful interaction and exploration novelty, enabling generalization to previously unseen interface states.

- We introduce a curiosity mechanism that leverages a world model to estimate state novelty in transition dynamics. This encourages the agent to actively explore diverse states, addressing the challenge of sparse supervision in open-ended GUI environments.

- We develop an experience stream distillation pipeline, where each generation's exploration experience is reused to fine-tune future agents. This strategy improves exploration efficiency, reduces reliance on manually curated datasets, and enables continuous capability evolution.

## 2 RELATED WORK

### 2.1 OPEN-WORLD EXPLORATION

Open-world or open-ended environments are characterized by vast state spaces, variable objectives, and sparse rewards, where agents must engage in active exploration to acquire rewards and accomplish tasks. Researchers have focused on enhancing RL agents' exploration capabilities through various approaches, including intrinsic motivation (such as curiosity-driven rewards) Burda et al. (2018); Zhang et al. (2021a), unsupervised skill discovery Eysenbach et al. (2018); Sharma et al. (2020); Wang et al. (2023), goal-oriented policy learning Campero et al. (2021), and exploration rewards Li et al. (2025). Random Network Distillation (RND) Burda et al. (2018) efficiently drives agent exploration to novel states by utilizing the prediction error between a fixed random target network and a trainable predictor network as an intrinsic reward signal. These methods aim to enable agents to autonomously discover meaningful behavioral patterns in the absence of external rewards, thereby achieving broader skill sets and higher sample efficiency in complex environments. In Li et al. (2025), the authors tackle the open-world 3D environment of Minecraft by first generating simulated trajectories that progressively zoom in on a target object, then use a task reward model to score these image sequences and train a world model to predict environmental dynamics. This world model produces an affordance map indicating feasible regions for the target object, which is then used as an intrinsic reward to drive the agent to approach the object and complete the task. In our work, inspired by Burda et al. (2018) and Li et al. (2025), we introduce a world model into the exploration process to approximate environmental state transitions. We use the discrepancy between the world model's predictions and actual post-action states as an intrinsic exploration reward, driving the model to interact effectively with the environment and reach unexplored states.

## 2.2 DIVERSE EXPLORATION BY LM

For statically deployed models, various decoding strategies can be employed to control output diversity, such as adjusting temperature sampling or top-p decoding parameters. Additionally, researchers have explored using Monte Carlo Tree Search (MCTS) to generate diverse solution paths, followed by validation through external tools or Process-supervised Reward Models (PRM) to verify path correctness, thereby enhancing the model's capability to solve practical problems Xin et al. (2024); Lightman et al. (2023); Qi et al. (2024). In Pan et al. (2025), researchers evaluate LLMs' exploration capabilities through the Little Alchemy 2 game, revealing that most LLMs—with the exception of the OpenAI o1 model—performed worse than humans, highlighting the need to improve LLMs' ability to explore open environments. In Dou et al. (2025), researchers find that LLMs' early promising solutions are often forgotten during RL fine-tuning due to policy gradient updates. Their proposed RRL algorithm stores and replays valuable early trajectories, enabling models to revisit previous approaches as their capabilities grow. This highlights the value of sustained exploration in RL-based LLM optimization. Another research Wang et al. (2025b) shows that training LLMs by RL can reduce exploration as models tend to repeat early high-reward responses, leading to repetitive outputs rather than exploring diverse solutions.

## 2.3 COMPUTER USE AGENTS

The GUI environment provides a natural testbed for training LLM/VLM agents, combining textual and visual elements to leverage large models' multimodal capabilities. Recent work explores end-to-end training of VLMs for controlling desktops Zhang et al. (2024); Lin et al. (2024), mobile phones Wu et al. (2024a); Wang et al. (2024), browsers Deng et al. (2023), and games Tan et al. (2024a;b), with commercial systems such as OpenAI Operator[1], Claude Computer Use[2] and Manus[3] also demonstrating practical deployment. RL-based approaches further enhance adaptability: DigiRL Bai et al. (2024) introduces an offline-to-online pipeline, while DistRL Wang et al. (2025a) scales up RL training for Android tasks. More recent efforts emphasize reasoning and generalization—InfiGUI-R1 Liu et al. integrates spatial reasoning via a two-stage Actor2Reasoner framework, UI-R1 Lu et al. applies rule-based reinforcement fine-tuning for efficient GUI action prediction, and GUI-R1 Luo et al. extends this paradigm across platforms with unified rule modeling. Unlike these offline-data-driven approaches, our work targets open online environments, using heuristic-based rewards to cost-effectively collect large-scale streams of GUI interaction experiences.

## 3 FRAMEWORK

In this section, we present the overall framework of our ScreenExplorer model, which is designed to train Vision-Language Model (VLM) agents to perform autonomous and exploratory interactions within graphical user interface (GUI) environments. To support this, we construct a GUI-based operating system as a RL environment, where the VLM agent interacts with GUI in a human-like manner by outputting function calls for mouse and keyboard operations. Technical details of this environment are provided in Appendix A. We formulate GUI exploration as a Markov Decision Process (MDP), where the agent perceives the environment through visual and textual signals, and outputs structured actions and intent descriptions using a VLM-based policy. The framework integrates several core components: a reward system tailored for encouraging exploration and meaningful interaction, a world model that predicts state transitions to support curiosity-driven behavior, and a training pipeline that combines RL with supervised distillation. The experience stream distillation allows the model to efficiently inherit and build upon past exploration knowledge, improving both exploration breadth and behavioral robustness. The subsequent subsections detail the MDP formulation, reward design, world model learning, and policy optimization with GRPO.

---

[1] https://openai.com/index/introducing-operator/
[2] https://www.anthropic.com/news/3-5-models-and-computer-use
[3] https://manus.im/

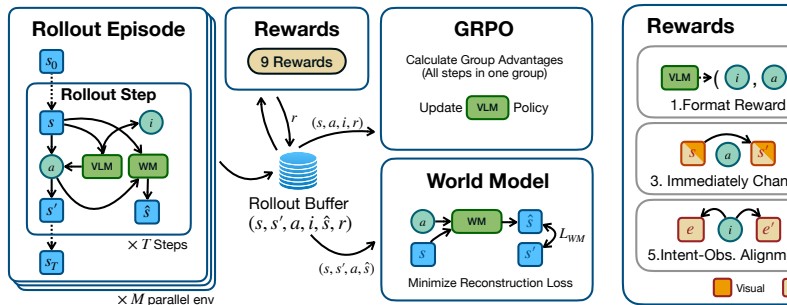

(a) One episode rollout collection and training progress.  (b) Reward function.

Figure 2: Framework overview: (a)We run $M$ parallel environments for $T$ steps per episode. At each step, the VLM takes state $s$ and outputs an intent $i$ and action $a$, the environment returns the post-action state $s'$, and the world model predicts the next state $\hat{s}$. All transitions are stored in a rollout buffer, where a reward function computes an exploration reward for each action. The VLM is then updated via GRPO, while the world model learns transitions by minimizing reconstruction error; (b)The reward function consists of nine terms that enforce correct action formatting, encourage large state changes, and align intents with observed states.

## 3.1 Modeling Exploration in GUI as a Markov Decision Process (MDP)

We model GUI exploration as an MDP with state $s = (o, e)$, where $o$ is the screenshot and $e$ is the OCR text embedding. A Vision-Language Model (VLM) serves as the policy $\pi_\theta$, which, given $s$, outputs an action $a$ (as a textual function call) and an intent $i$: $(a, i) = \pi_\theta(s)$. Executing $a$ yields a new state $s'$, while a world model predicts $\hat{s}$.

Each step produces a tuple $(s, a, i, s', \hat{s}, r)$ stored in the rollout buffer $\mathcal{B}$. An episode starts from $s_0$ (initial desktop) and proceeds for $T$ steps to $s_T$. With $N$ parallel environments, $N \times T$ such tuples are collected to update $\pi_\theta$ (e.g., via GRPO) and train the world model. Details of the optimization procedure are provided in Appendix C.

## 3.2 Reward Function

To encourage the agent's effective interaction with the environment while promoting exploration of unseen states, we design a set of reward functions. Effective interaction requires the model to output correct actions that induce state changes in the environment. In GUI environments, this means the Vision-Language Model (VLM) must output actions in the proper format, and the execution of the action should result in substantial changes to the screen's visual or textual content. Specifically, our designed rewards include the following categories:

- **Format Reward** ($r_{format}$): At each action step, the environment verifies whether the model's output is in the correct format.

  i) If the format is correct, the reward $r_{format} \leftarrow 1$.

  ii) If the format is incorrect, the reward $r_{format} \leftarrow 0$, and the action $a$ is set to a null action.

- **Exploration Reward:** To encourage the agent to explore novel environment states, we design the exploration reward. We use cosine similarity, denoted as $sim(\cdot, \cdot)$, to measure the similarity between states (e.g., visual and textual representations). Lower similarity between states indicates higher exploration diversity. For action $a_t$, the exploration reward is composed of the following terms:

  i) *Instantaneous Change Reward* ($r_{inst}^{vis}$ & $r_{inst}^{text}$): Measures the degree of state change (e.g., visual and textual representations) immediately before and after the action: $r_{inst}^{vis}(t) := 1 - sim(o_t, o_t')$, $r_{inst}^{text}(t) := 1 - sim(e_t, e_t')$. Where $o_t, e_t$ are the visual and textual representations before the action, and $o_t', e_t'$ are those after the action.

  ii) *Subsequent Change Reward* ($r_{seq}^{vis}$ & $r_{seq}^{text}$): To measure the diversity gain brought by action $a_t$ to the overall state sequence, we evaluate the average similarity between all state pairs before and

after the action occurs. The difference from this average similarity is attributed to the action's subsequent change reward, encouraging the agent to take actions that lead to novel long-term state sequences: $r_{seq}^{vis}(t) := E_{i<t,j>t}[1 - sim(o_i', o_j')]$, $r_{seq}^{text}(t) := E_{i<t,j>t}[1 - sim(e_i', e_j')]$. Where $E_{i<t,j>t}[\cdot]$ denotes the expectation of similarity calculated over all state pairs with $i < t$ and $j > t$.

iii) *World Model Curiosity Reward ($r_{world}^{vis}$ & $r_{world}^{text}$):* Captures the agent's "surprise" by comparing the actual post-action states with the states predicted by a world model, thereby encouraging the agent to explore uncertain or difficult-to-predict areas of the environment: $r_{world}^{vis}(t) := 1 - sim(o_t', \hat{o}_t)$, $r_{world}^{text}(t) := 1 - sim(e_t', \hat{e}_t)$. Where $\hat{o}_t, \hat{e}_t$ are the visual and textual states predicted by the world model after the action.

- **Intent-State Alignment Rewards:** To encourage the model to observe the environment more carefully and generate intents (or explanations) related to its actions or the environment state, we design two intent-state alignment rewards. These rewards are computed by comparing the text embeddings of the agent's output intent string $i_t$ with text embeddings obtained from screen OCR:

  i) *Environment Description Reward ($r_{des}$):* Encourages the agent's intent string $i_t$ to be related to the overall environment content by calculating similarity with pre- and post-action text $e_t, e_t'$: $r_{des}(t) := sim(i_t, e_t) + sim(i_t, e_t')$.

  ii) *Intent Interpretation Reward ($r_{inter}$):* Encourages the agent's intent string $i_t$ to be related to the specific UI element located at the action coordinates. We extract the text $e_{box}$ from the OCR box where the action coordinates are located and compute its similarity with the intent text: $r_{inter}(t) := sim(i_t, e_{box})$. Where $e_{box}$ denotes the text embedding generated from the text within the OCR box containing the action coordinates.

**Overall Reward:** We combine the above reward terms to construct the final reward function as:

$$r := r_{format} \times (r_{inst}^{vis} + r_{inst}^{text} + r_{seq}^{vis} + r_{seq}^{text} + r_{world}^{vis} + r_{world}^{text} + r_{des} + r_{inter}). \quad (1)$$

Here, $r_{format}, r_{inst}^{vis}, \ldots$ are used as shorthand for $r_{format}(t), r_{inst}^{vis}(t), \ldots$ to simplify notation. This formula indicates that the sum of other reward terms is included in the total reward only when the action format is correct ($r_{format} = 1$); otherwise ($r_{format} = 0$), the total reward is zero.

## 3.3 LEARNING STATE TRANSITIONS WITH A WORLD MODEL

The world model $\mathcal{M}_\phi : \mathcal{S} \times \mathcal{A} \to \mathcal{S}$ predicts the next state given the current state and action: $\hat{s} \leftarrow \mathcal{M}_\phi(s, a)$. The action's word tokens, image's visual tokens and text embedding $e$ are concatenated to form the input sequence for the world model, illustrate as, $\hat{s} := (\hat{o}, \hat{e}) = \mathcal{M}_\phi(o, e, a)$. The world model parameters $\phi$ are optimized to minimize the reconstruction loss between predicted and actual next states as follows. More details of the world model can be found in Appendix B.

$$\mathcal{L}_{WM}(\phi) = \mathbb{E}_{(s,a,s') \sim \mathcal{B}} \left[ \|\mathcal{M}_\phi(s, a) - s'\|^2 \right] \quad (2)$$
$$= \mathbb{E}_{(o,e,a,o',e') \sim \mathcal{B}} \left[ \|\hat{o} - o'\|^2 + \|\hat{e} - e'\|^2 \right].$$

## 3.4 TRAINING VLM AGENT WITH GRPO

We optimize VLM Agent policy by Group Relative Policy Optimization (GRPO)Shao et al. (2024). All samples in the buffer are treated as a GRPO group and the advantage of each action $A_i$ is estimated as: $A_i = (r_i - \text{mean}(\{r_j\}_{j=1}^{|\mathcal{B}|}))/\text{std}(\{r_j\}_{j=1}^{|\mathcal{B}|})$.

GRPO optimizes the policy model by maximizing the following objective function:

$$\mathcal{J}_{GRPO}(\theta) = \mathbb{E} \left[ (o, a, r) \sim \mathcal{B}, \{a_i\}_{i=1}^{|\mathcal{B}|} \sim \pi_{\theta_{old}}(a \mid o) \right]$$

$$\frac{1}{|\mathcal{B}|} \sum_{i=1}^{\mathcal{B}} \left\{ \min \left[ \frac{\pi_\theta(a_i \mid o_i)}{\pi_{\theta_{old}}(a_i \mid o_i)} A_i, \text{clip} \left( \frac{\pi_\theta(a_i \mid o_i)}{\pi_{\theta_{old}}(a_i \mid o_i)}, 1 - \varepsilon_{low}, 1 + \varepsilon_{high} \right) A_i \right] - \beta \mathbb{D}_{KL}(\pi_\theta \| \pi_{ref}) \right\}, \quad (3)$$

where $\epsilon_{low}$ and $\epsilon_{high}$ controls the clip range; $\mathbb{D}_{KL}(\pi_\theta \| \pi_{ref})$ measures the KL divergence between the current policy $\pi_\theta$ and a reference policy $\pi_{ref}$. $\beta$ weights the KL divergence regularization term,

which prevents the optimization process from deviating too far from the original policy:

$$\mathbb{D}_{KL}\left(\pi_\theta \| \pi_{ref}\right) = \frac{\pi_{ref}\left(a_i \mid o_i\right)}{\pi_\theta\left(a_i \mid o_i\right)} - \log \frac{\pi_{ref}\left(a_i \mid o_i\right)}{\pi_\theta\left(a_i \mid o_i\right)} - 1. \tag{4}$$

In the RL phase, starting from the base model, the agent learns to interact with the environment, explores it diversely, and collects trajectories to form a data set of experience stream. Next, these streams are passed through a quality and diversity filter, which selects the most informative and diverse trajectories. In the distillation phase, we reinitialize from the base model and perform supervised fine-tuning (SFT) using the filtered data to distill a new model. Through supervised fine-tuning, the experience gained from RL training can be efficiently transferred to the distilled model, avoiding the need to learn from scratch. Our training approach, which combines RL and distillation, gradually enhances the capacity of VLM agents to perform meaningful, diverse actions in complex GUI environments. More details of the distillation process are provided in Appendix D.

## 4 EXPERIMENT

We constructed a Linux-based desktop operating system as our environment, which includes basic software such as web browsers and the LibreOffice suite. The VLM Agent can interact with the environment by outputting action commands in JSON format. The task prompt is fixed as "Your goal is to explore this environment as much as possible within a limited number of steps", thereby encouraging the model to engage in diverse exploration within the environment. Details of the environment configuration and the complete prompt can be found in Appendix A.

### 4.1 EVALUATION METRICS ON EXPLORATION DIVERSITY

We evaluate the performance of agents' diverse exploration in GUI environments by measuring both the diversity of environmental states within individual trajectories and the diversity of states across all trajectories within a group. We use the cosine similarity $sim(\cdot, \cdot)$ to measure the similarity between environmental states, as defined in subsection 3.2.

**Trajectory-level Diversity:** For a trajectory $\tau$ consists of $T$ states $\{s_1, \ldots, s_T\}$, the corresponding visual representations are $\{o_1, \ldots, o_T\}$, and textual representations are $\{e_1, \ldots, e_T\}$, define the visual and textual sequence diversity of trajectory $\tau$ as:

$$d_{\text{seq}}^{\text{vis}}(\tau) = \frac{1}{T(T-1)} \sum_{1 \le k < l \le T} \left[1 - sim(o_k', o_l')\right]; d_{\text{seq}}^{\text{text}}(\tau) = \frac{1}{T(T-1)} \sum_{1 \le k < l \le T} \left[1 - sim(e_k', e_l')\right]. \tag{5}$$

**Group-level Diversity:** For a group of trajectories $\mathcal{G} = \{\tau^1, \ldots, \tau^M\}$, each trajectory $\tau$ consists of $T$ states. We flatten all $M$ trajectories into a set of $N = M \times T$ different states and denote their visual and textual embeddings by $\{(o_k, e_k)\}_{k=1}^N$, define the visual and textual diversity of group $\mathcal{G}$ as:

$$D_{\text{grp}}^{\text{vis}}(\mathcal{G}) = \frac{1}{N(N-1)} \sum_{1 \le k < l \le N} \left[1 - sim(o_k', o_l')\right]; D_{\text{grp}}^{\text{text}}(\mathcal{G}) = \frac{1}{N(N-1)} \sum_{1 \le k < l \le N} \left[1 - sim(e_k', e_l')\right]. \tag{6}$$

### 4.2 EXPLORATION PERFORMANCE

**Baselines and Evaluation Settings.** We adopt *Qwen2.5-VL-7B* and *Qwen2.5-VL-3B* as the backbone of ScreenExplorer. All baselines are categorized into two groups: *General Used Models* and *GUI-specific models*, which are accessible via API or static deployment. Our model includes variants: *ScreenExplorer-3B*, *ScreenExplorer-7B* and *ScreenExplorer-3B-Distill*. *ScreenExplorer-3B-Distill* refers to the model after experience stream distillation. Our variant model is statically deployed under identical experimental conditions as baselines, we selected the best-performing checkpoint for evaluation. Except for OpenAI Computer Use, all other models are tested under two sampling temperatures, $t = 1.0$ and $t = 0.5$. More details regarding prompts and other settings are provided in Appendix E.

Table 1: Overall Performance in Correct Formatting and Exploration. box indicates the lowest scores, while **bold** text denotes the highest scores.

| Model | Setting | Correct Format | Trajectory-level | | Group-level | | Avg. Diversity |
|---|---|---|---|---|---|---|---|
| | | | $d_{\text{seq}}^{\text{vis}}(\tau)$ | $d_{\text{seq}}^{\text{text}}(\tau)$ | $D_{\text{grp}}^{\text{vis}}(\mathcal{G})$ | $D_{\text{grp}}^{\text{text}}(\mathcal{G})$ | |
| *General Used Models* | | | | | | | |
| OpenAI gpt-4o | $t = 1.0$ | 0.95 | 0.25 | 0.16 | 0.35 | 0.25 | 0.25 |
| | $t = 0.5$ | **1.00** | 0.13 | 0.17 | 0.18 | 0.26 | 0.18 |
| Qwen2.5-VL-72B | $t = 1.0$ | 0.96 | 0.39 | 0.26 | 0.69 | 0.39 | 0.43 |
| | $t = 0.5$ | **1.00** | 0.23 | 0.16 | 0.39 | 0.22 | 0.25 |
| Qwen2.5-VL-7B | $t = 1.0$ | 0.68 | 0.44 | 0.26 | 0.57 | 0.37 | 0.41 |
| | $t = 0.5$ | 0.75 | 0.37 | 0.21 | 0.34 | 0.29 | 0.32 |
| Qwen2.5-VL-3B | $t = 1.0$ | 0.62 | 0.16 | 0.10 | 0.40 | 0.19 | 0.21 |
| | $t = 0.5$ | 0.84 | 0.15 | 0.08 | 0.31 | 0.14 | 0.17 |
| *GUI-specific Models* | | | | | | | |
| OpenAI Computer Use | default | 0.95 | 0.28 | 0.21 | 0.60 | 0.32 | 0.35 |
| doubao-1.5-ui-tars | $t = 1.0$ | 0.82 | 0.41 | 0.24 | **0.76** | 0.38 | 0.45 |
| | $t = 0.5$ | 0.75 | 0.30 | 0.17 | 0.64 | 0.32 | 0.36 |
| *Ours* | | | | | | | |
| ScreenExplorer-3B | $t = 1.0$ | 0.99 | 0.57 | 0.33 | 0.68 | 0.45 | 0.51 |
| | $t = 0.5$ | **1.00** | 0.57 | 0.33 | 0.72 | 0.46 | 0.52 |
| ScreenExplorer-7B | $t = 1.0$ | 0.98 | 0.65 | 0.42 | 0.64 | 0.45 | 0.54 |
| | $t = 0.5$ | 0.995 | 0.64 | **0.44** | 0.64 | **0.49** | **0.55** |
| ScreenExplorer-3B-Distill | $t = 1.0$ | 0.93 | 0.64 | 0.37 | 0.68 | 0.43 | 0.53 |
| | $t = 0.5$ | 0.99 | **0.66** | 0.41 | 0.67 | 0.44 | **0.55** |

**Overall Results.** Table 1 summarizes the results across all baselines and our models. We observe that *ScreenExplorer* consistently outperforms both general-purpose and GUI-specific models on all exploration diversity metrics.

Among general-purpose models, *gpt-4o* achieves strong scene understanding but suffers from inaccurate coordinate localization, resulting in repeated failed actions due to its static deployment. Within the Qwen2.5-VL family, the 72B variant demonstrates the strongest performance, while the *Qwen2.5-VL-3B* version performs the worst. The *Qwen2.5-VL-7B* variant yields better exploration diversity than 3B and approaches 72B, though it remains limited in correct formatting compared to larger-scale models.

In the category of GUI-specific models, *OpenAI Computer Use* and *doubao-1.5-ui-tars* are able to localize UI elements precisely and perform responsive operations, leading to competitive sequence- and group-level diversity. Nevertheless, they fall short of the exploration breadth achieved by our RL-trained models.

Our models achieve near-perfect format accuracy while substantially improving exploration diversity. Specifically, *ScreenExplorer-7B* attains the best overall performance, reaching the highest textual sequence diversity $(0.44)$ and group-level textual diversity $(0.49)$, with an average diversity of $0.55$. *ScreenExplorer-3B* improves format correctness from 0.62/0.84 to 0.99/1.00 (at $t = 1.0/0.5$) through RL training.

We also note that sampling temperature strongly influences the trade-off between correctness and diversity. Higher temperatures $(t = 1.0)$ generally encourage more diverse actions and trajectories, while lower temperatures $(t = 0.5)$ favor stricter instruction-following and correct formatting. This observation aligns with prior findings on diversity–fidelity trade-offs in decoding strategies.

**Distillation Results.** We trained the *ScreenExplorer-3B-Distill* version by distilling the experience streams of *ScreenExplorer-3B* into the original *Qwen2.5-VL-3B* backbone. *ScreenExplorer-3B-Distill* achieves exploration performance on par with *ScreenExplorer-3B*, which demonstrate that experience stream distillation can directly transfer the environmental interaction and exploration capabilities of the previous generation into the student model.

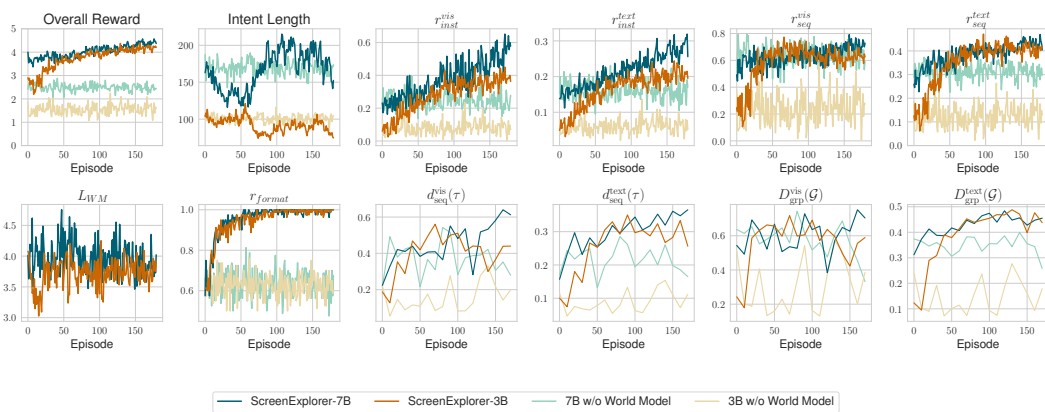

Figure 3: The reward dynamics and performance metrics of *ScreenExplorer-7B* and *ScreenExplorer-3B* during the RL training process.

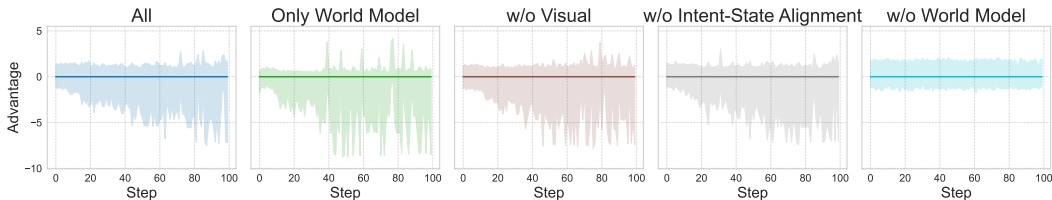

Figure 4: Advantage Range. By introducing the world model reward increases the variance of GRPO advantage, enabling smoother cold-start optimization and more effective exploration through better gradient differentiation across sample groups.

### 4.3 TRAINING DYNAMICS

To better understand the progression of model's different capabilities, we analyze the key metrics in RL training process. As illustrated in Figure 3, all rewards from *ScreenExplorer-7B* and *ScreenExplorer-3B* increase as training progresses, leading to a gradual improvement in overall reward. In the initial training phase, the Format Reward shows rapid improvement first. Subsequently, four exploration diversity rewards begin to rise, indicating that the model gradually learns to interact effectively with the environment and explore deeper into pages. After about 100 steps, visual and textual exploration diversity rewards gradually saturate, leading the model to enhance its intent outputs to achieve higher intent interpretation rewards. Throughout the training process, the world model loss $\mathcal{L}_{WM}$ first slightly decreases, then oscillates at a high value, indicating that the world model maintains a consistently high level of curiosity.

**Why World Model?** To evaluate the effectiveness of the world model, we conducted two ablation experiments by removing the curiosity reward from the world model, denoted as *7B w/o world model* and *3B w/o world model*. As shown in Figure 3, in the ablation groups, the policy network struggles to improve format rewards and establish effective interaction with the environment. In contrast, both experiments incorporating the world model successfully overcome the cold start phase, as evidenced by continuous improvements in both reward and diversity metrics.

Furthermore, we analyzed the GRPO advantage values of both the ablation and full-rewards groups, illustrated in Figure 4. Adding curiosity rewards from the world model increased advantage variance, which smooths the score gradients within a group of samples, enabling the RL process to more rapidly identify optimization directions in the cold start phase of exploration. Similar observations were also noted in Razin et al. (2025), which argues that an effective reward function not only needs to be accurate but also requires sufficient variance. For additional ablation studies on other reward components, refer to Appendix G.

| Episode | $s_0$ | $s_1'$ | $s_2'$ | $s_3'$ | $s_4'$ | $s_5'$ | $s_6'$ | $s_7'$ | $s_8'$ | $s_9'$ |
|---|---|---|---|---|---|---|---|---|---|---|
| 0 | | | | | | | | | | |
| 50 | | | | | | | | | | |
| 90 | | | | | | | | | | |
| 100 | | | | | | | | | | |

Figure 5: Examples of Trajectories from *ScreenExplorer-3B*. Through RL training, the model developed increasingly effective interactions with the environment, enabling exploration of deeper pages.

**Case Study** Figure 5 shows training trajectories that demonstrate how the model developed effective environmental interactions and explored deeper pages through RL. Our case study demonstrates that as reinforcement learning training advances, the model develops diverse exploration strategies. These exploratory capabilities enable the model to better adapt and generalize in open-world environments. Additionally, the model can extract action intent labels from the intent field without manual annotation, offering a scalable method for building large-scale datasets in the future. Additional case studies are presented in Appendix H.

## 5 DISCUSSION

**Capability Compositionality.** Small-scale VLMs possess basic capabilities like OCR, computer knowledge, and visual detection/localization, but struggle to combine these effectively. In Appendix F, our cases show these models often fail to coordinate their skills, with conflicts arising between language generation and visual processing abilities. Through training in GUI environment, models can combine these inherent capabilities into higher-level exploration skills;

**Noisy TV Problem and Exploration Dilemma.** Curiosity-driven exploration in RL faces the "Noisy TV problem." Burda et al. (2018); Mavor-Parker et al. (2024) We find that agents get stuck on irrelevant, perpetually novel stimuli (e.g., clicking the first news item or opening video sites). We found long period of training LLMs with RL can reduce exploration, as they tend to repeat early high-reward responses, leading to exploration traps. To improve exploration diversity and prevent stagnation, experience stream distillation was implemented.

**Limitations.** We mainly use screenshots and text content to measure environmental state similarity. However, structured state information from operating system, such as process running status and variables, are also suitable choices. But, achieving this requires a considerable amount of systems engineering work. Although we introduced vLLM Kwon et al. (2023) to accelerate VLM decoding, the time cost of sampling and training the VLM agent in a real GUI environment is prohibitively high, which limits our ability to conduct larger-scale experiments. We plan to implement distributed sampling and training methods in the future to speed up the learning process.

## 6 CONCLUSION

In this work, we identified a critical limitation of current GUI agents: their inability to generalize beyond their training data and their heavy reliance on large-scale, human-curated datasets. To bridge this gap, we introduced ScreenExplorer, a novel VLM-based agent grounded in a paradigm of autonomous exploration. We first design an intrinsic motivation mechanism that combines a world model-inspired curiosity reward with state-change-based signals. This combination effectively guides the agent through the initial cold-start phase and incentivizes deep exploration of novel states. Coupled with experience stream distillation for continual learning, our agent achieves remarkable generalization in unseen applications, significantly outperforming static baselines. This work demonstrates that exploration-driven learning is a scalable and effective alternative to static datasets, paving the way for adaptive, self-improving GUI agents.

## USE OF LLMs

- The ideas, methods, experimental designs, and writing structure of this paper were independently developed by the authors. LLMs were used for translation, phrasing refinement, and grammatical corrections, without contributing to the generation of viewpoints, methodologies, or conclusions.

- During code development, GitHub Copilot was used to assist with code completion and refactoring; however, all key implementations, algorithm designs, and experimental pipelines were written, reviewed, and validated by the authors through reproduction experiments.

- The experimental data, training logs, and figures presented in the paper are derived from actual runs and statistics, containing no data generated or fabricated by LLMs.

- For revision suggestions provided by LLMs, the authors reviewed each recommendation individually and made adjustments when necessary to ensure accurate representation aligned with the authors' intentions.

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

# A    TECHNICAL DETAILS OF GUI ENVIRONMENT AND VLM AGENT

Our GUI environment is constructed on a Linux desktop system and wrapped as a Gymnasium [4] environment for reinforcement learning training. The observation space consists of RGB screenshots with configurable resolution. The action space is structured as a dictionary space, supporting a variety of mouse and keyboard interactions, with detailed action types and examples shown in Table 2.

Table 2: GUI Environment Action Space

|  | Action Type | Attributes | Action Function Call Example |
|---|---|---|---|
|  | Move | (x:int, y:int) | `Move(960, 540)` |
|  | Click | (x:int, y:int) | `Click(960, 540)` |
|  | Right Click | (x:int, y:int) | `RightClick(960, 540)` |
| Mouse | Double Click | (x:int, y:int) | `DoubleClick(960, 540)` |
|  | Scroll Up | (x:int, y:int) | `ScrollUp(960, 540)` |
|  | Scroll Down | (x:int, y:int) | `ScrollDown(960, 540)` |
|  | Drag To | (x:int, y:int) | `DragTo(960, 540)` |
| Keyboard | Key or Combined-keys | (key:string) | `Key("Space") Key("Shift+K")` |
|  | Text | (x:int, y:int, text:string) | `Text(960, 540, "Hello World!")` |
|  | None | - | `None()` |

The environment validates the action strings VLM agent's output, verifying whether they conform to valid function call formats and whether the attributes values fall within acceptable ranges. If the format is incorrect, the action type will be set to None. Additionally, the environment invokes an OCR module to parse both pre-action and post-action screenshots, which provides input for text embedding generation in the world model, reward computation, and metric calculation. The GUI environment in this paper employs the configurations specified in Table 3.

Table 3: Configuration of GUI Environment

| Environment Configuration | |
|---|---|
| Screen width | 1920 |
| Screen height | 1080 |
| Wait after action execution | 1.0 seconds |
| Maximum steps per episode | 10 |
| Parallel environments | 8 |
| OCR model | PP-OCRv4-mobile-det + PP-OCRv4-mobile-rec |

We employ the Qwen2.5-VL series as our base VLM, where the input prompts consist of a fixed textual prompt and a dynamically changing image of current screenshot. The screenshots maintain their original resolution. The following presents the textual components of the prompt:

```
You are exploring a computer desktop environment with a screen
size of {{video_width}}x{{video_height}}. You can interact with
it using the keyboard and mouse. Your goal is to explore this
environment as much as possible within a limited number of steps.

Available action format:
- Move(x, y): Move the mouse to coordinates (x, y)
- Click(x, y): Left-click at coordinates (x, y)
- RightClick(x, y): Right-click at coordinates (x, y)
- DoubleClick(x, y): Double left-click at coordinates (x, y)
- ScrollUp(x, y): Scroll up at coordinates (x, y)
- ScrollDown(x, y): Scroll down at coordinates (x, y)
- Text(x, y, "text"): Enter text "text" at coordinates (x, y)
- Key("key"): Press a single key
```

---

[4] https://gymnasium.farama.org/

```
- Key("Shift+K"): Combination key

Note that opening icons on the desktop requires a double click.
Please select a meaningful action to continue exploring. Each
action consumes steps, so please choose the most valuable
operation.

Please reply in the following JSON format:

{
    "intent": "Explanation of why this action was chosen and what
    goal it aims to achieve",
    "action": "Specific action, for example Click(123, 456)"
}
```

We utilize vLLM's Structured Outputs feature[5] to enforce model outputs as JSON strings containing "intent" and "action" fields. After updating the VLM weights in each episode, the parameters are immediately synchronized to vLLM before beginning sampling for the next episode, maintaining an online training paradigm.

## B    TECHNICAL DETAILS OF WORLD MODEL

We employ a LLaMA-style Transformer as the backbone of our world model, the architectural design is illustrated in Figure 6. The model predicts the next environment states by given the current action and state.

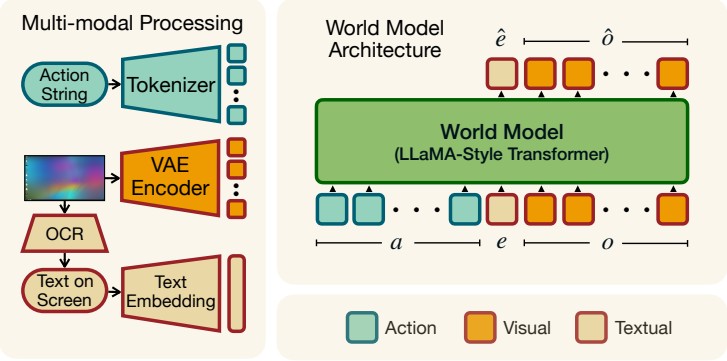

Figure 6: World Model Architecture

For action $a$, we use the LLaMA tokenizer to encode the action's function-call-style string into word tokens. For all text on the screen extracted by the OCR module, we use a pre-trained text embedding model to project all text into one dense vector, denoted as $e$. For image $o$, we utilize a pre-trained VAE encoder to map the screenshot to visual tokens. The world model is trained with state reconstruction loss, which minimizes the mean squared error between predicted and target image tokens and text embeddings.

## C    TRAINING PROCESS

Algorithm 1 outlines the RL training procedure. Table 4 lists the VLM training configurations and hyperparameters used for GRPO, and Table 5 summarizes the world model's training settings. All experiment was conducted using one Nvidia A100 GPU. The *ScreenExplorer-3B* model requiring approximately 26.8 hours to complete 200 steps, and *ScreenExplorer-7B* requiring approximately 25

---

[5]https://docs.vllm.ai/en/latest/features/structured_outputs.html

hours to complete 200 steps. The primary time consumption occurs in processes such as interacting with the environment and OCR parsing.

---

**Algorithm 1:** RL Training Process of ScreenExplorer

---

**Input:** Maximum episodes $N$, environment $\mathcal{E}$, actor model $\pi_\theta$, world model $f_\phi$, buffer $\mathcal{B}$
**Output:** Trained actor model $\pi_\theta$ and world model $f_\phi$
**for** *episode* $= 1$ **to** $N$ **do**
    Initialize rollout buffer $\mathcal{B}$;
    Reset environment $\mathcal{E}$ to obtain initial observation $s_0$;
    // Collect rollouts
    **for** *step* $t = 0$ **to** *max_steps* **do**
        Select action by VLM agent: $a_t \sim \pi_\theta(a|o_t)$;
        Execute $a_t$ in $\mathcal{E}$, obtain $s'_t$ and $r_{format}$ ;
        World model predicts next state: $\hat{s}_t = f_\phi(s_t, a_t)$ ;
        Store $(s_t, a_t, s'_t, \hat{s}_t, r_{format})$ in $\mathcal{B}$;
    // Reward and advantage computation
    **foreach** *trajectory* $\tau$ *in* $\mathcal{B}$ **do**
        Compute overall reward $r$ based on trajectory $\tau$, store in $\mathcal{B}$.
    Compute GRPO advantages for all samples in $\mathcal{B}$;
    // World model training
    **for** *epoch* $= 1$ **to** *world_model_training_epoch* **do**
        **foreach** *batch in* $\mathcal{B}$ **do**
            Update $\phi$ to minimize reconstruction loss $\mathcal{L}_{\text{WM}}(\phi)$ ;
    // VLM optimization
    **foreach** *batch in* $\mathcal{B}$ **do**
        Update actor model $\pi_\theta$ using GRPO with computed advantages;

---

Table 4: VLM Training Configuration and GRPO Hyperparameters

| VLM | |
|---|---|
| Batch size | 16 |
| Mixed precision | bf16 |
| Maximum gradient norm | 1.0 |
| LoRA rank | 16 |
| Learning rate | 4e-5 |
| Training batch size | 16 |
| Maximum completion length | 128 |
| Rollout temperature | 1.0 |
| **GRPO Parameters** | |
| KL divergence coefficient($\beta$) | 0.04 |
| PPO lower bound($\varepsilon_{low}$) | 0.2 |
| PPO upper bound($\varepsilon_{high}$) | 0.28 |

Table 5: World Model Training Configuration

| World Model | |
|---|---|
| World model base | Llama-3.2-1B |
| Image tokenizer model | Cosmos-Tokenizer-CI16x16 |
| Text embedding model | BAAI/bge-m3 |
| Training epoch | 3 |
| Training batch size | 32 |
| Learning rate | 4e-5 |
| Mixed precision | bf16 |
| Maximum gradient norm | 1.0 |

## D EXPERIENCE STREAM DISTILLATION

The purpose of Experience Stream Distillation is to identify valuable exploratory behaviors from historical exploration trajectories generated during RL training. The model distilled using these actions can directly inherit the effective environmental interaction capabilities from its predecessor generation, while maintaining diversity in exploration capabilities through data filtering and balancing.

We first filter the trajectories generated during RL training, identifying and retaining diverse exploration steps that successfully complete specific tasks. These single-step actions are then organized into datasets for Supervised Fine-Tuning (SFT) on the base model. Figure 7(a) illustrates this process.

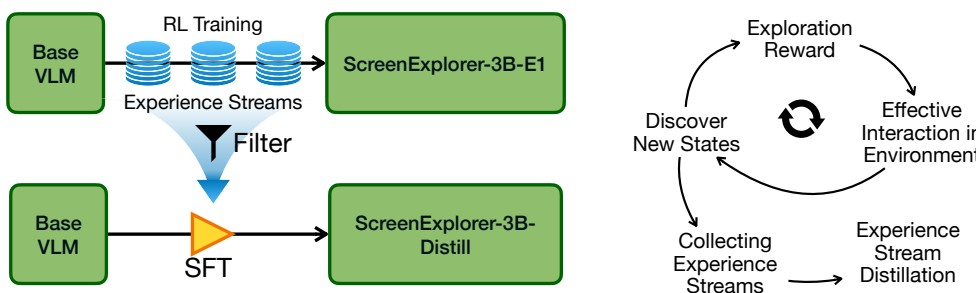

(a) Experience stream collection through RL training and distillation for next-generation models

(b) Data cycle for sustainable, self-fueling learning paradigm.

Figure 7: Training Pipeline Integrating RL and Experience Stream Distillation.

We employed two filtering strategies: manual filtering and automated filtering. For manual filtering of experience streams, we established the following criteria:

1. Begin from the 30-th episode.
2. The intent text description should clearly indicate a specific action (e.g., click, type, scroll) and specify a target (e.g., an icon, button, field, or screen location).
3. The executed action should accomplish the goal described in the intent.
4. The language in the intent should flow smoothly and contain no word repetitions.

In addition to manual filtering, we implemented an automated filtering process without human intervention, based on the following criteria:

1. Begin from the 30-th episode.
2. The output format is correct, where $r_{format} = 1$.
3. Advantages greater than $0$.
4. The intent text should clearly indicate a specific action, evaluated by *gpt-4o-mini-2024-07-18*.

The automated filtering employs the following prompts:

```
You are evaluating whether an intent string clearly specifies a
computer operation instruction.

A good intent should:
1. Clearly indicate a specific action (e.g., click, type, scroll)
2. Specify a target (e.g., an icon, button, field, or screen
location)
3. Be unambiguous about what the user wants to accomplish
4. The language flows smoothly and there are no words repeated.

Intent to evaluate: "{{intent}}"

If the intent meets the criteria above, rewrite it as a clear
task objective.
If the intent does not meet the criteria, mark it as not good and
use an empty string as the task.
Keep using the same language as the input intents.

Provide your evaluation in the following JSON format:
```

```json
{
  "is_good_intent": true/false,
  "reasoning": "Your detailed explanation for why this intent is good or not",
  "task": "Rewritten clear task objective if good, or empty string if not"
}
```

During both filtering processes, we did not introduce or construct additional data. Table 6 presents the exploration metrics of models after SFT using both manual filtering and automated filtering datasets. The results indicate that, compared to the baseline model, SFT training on either datasets effectively enhances the model's exploration capabilities and diversity. Compared to automatic filtering, manual filtering provides higher quality data for distillation by additionally verifying the consistency between intent descriptions and actually executed actions.

Table 6: Impact of Different Filtering Strategies in Experience Stream Distillation on Model Exploration Performance

| Model | # SFT Training Set | Setting | Correct Format | Trajectory-level | | Group-level | | Avg. Diversity |
|---|---|---|---|---|---|---|---|---|
| | | | | $d_{\text{seq}}^{\text{vis}}(\tau)$ | $d_{\text{seq}}^{\text{text}}(\tau)$ | $D_{\text{grp}}^{\text{vis}}(\mathcal{G})$ | $D_{\text{grp}}^{\text{text}}(\mathcal{G})$ | |
| *Base Model* | | | | | | | | |
| Qwen2.5-VL-3B | - | $t = 1.0$ | 0.62 | 0.16 | 0.10 | 0.40 | 0.19 | 0.21 |
| | | $t = 0.5$ | 0.84 | 0.15 | 0.08 | 0.31 | 0.14 | 0.17 |
| *Ours* | | | | | | | | |
| ScreenExplorer-3B (RL training from scratch) | - | $t = 1.0$ | 0.99 | 0.57 | 0.33 | 0.68 | 0.45 | 0.51 |
| | | $t = 0.5$ | 1.00 | 0.57 | 0.33 | 0.72 | 0.46 | 0.52 |
| ScreenExplorer-3B-Distill (Manual Filtering) | 216 | $t = 1.0$ | 0.93 | 0.64 | 0.37 | 0.68 | 0.43 | 0.53 |
| | | $t = 0.5$ | 0.99 | 0.66 | 0.41 | 0.67 | 0.44 | 0.55 |
| ScreenExplorer-3B-Distill (Automated Filtering) | 199 | $t = 1.0$ | 0.94 | 0.64 | 0.39 | 0.71 | 0.46 | 0.55 |
| | | $t = 0.5$ | 1.0 | 0.62 | 0.42 | 0.62 | 0.45 | 0.53 |

Figure 7(b) illustrates the data cycle process where an agent explores and collection experience stream data in an open-world environment, ultimately achieving continuous improvement through experience stream distillation. This data cycle mechanism enables the model to transcend the limitations of "static corpus + offline training," advancing toward a sustainable, self-fueling learning paradigm—representing a viable pathway for enhancing agent capabilities when human-generated data becomes exhausted in the future.

# E   BASELINE SETTINGS

For the General Used Models OpenAI gpt-4o and Qwen2.5-VL variants, we employed a fixed prompt designed to encourage free exploration. For OpenAI Computer Use, we utilize the officially recommended invocation method, with prompts that are not publicly disclosed. For doubao-1.5-ui-tars, we employ the following system prompts:

```
You are a GUI agent. You are given a task and your action history,
with screenshots. You need to perform the next action to complete
the task.
## Output Format
```
Thought: ...
Action: ...
```
## Action Space
click(start_box='[x1, y1, x2, y2]')
left_double(start_box='[x1, y1, x2, y2]')
right_single(start_box='[x1, y1, x2, y2]')
```

```
drag(start_box='[x1, y1, x2, y2]', end_box='[x3, y3, x4, y4]')
hotkey(key='')
type(content='') #If you want to submit your input, use "\n" at
the end of 'content'.
scroll(start_box='[x1, y1, x2, y2]', direction='down or up or
right or left')
wait() #Sleep for 5s and take a screenshot to check for any
changes.
finished(content='xxx') # Use escape characters \\', \\", and \\n
in content part to ensure we can parse the content in normal
python string format.
## Note
- Use Chinese in 'Thought' part.
- Write a small plan and finally summarize your next action (with
its target element) in one sentence in 'Thought' part.
## User Instruction
```

We employ the following task prompts:

```
Your goal is to explore this environment as much as possible
within a limited number of steps. Please select a meaningful
action to continue exploring. Note that opening icons on the
desktop requires a double click. You must only use mouse and
keyboard inputs. No other tools or input devices are permitted.
```

## F  ORIGINAL CAPACITY IN VLM

We find the *Qwen2.5-VL* model has demonstrated some fundamental capabilities in GUI exploration, as evidenced by the following representative cases in Figure 8 to 11.

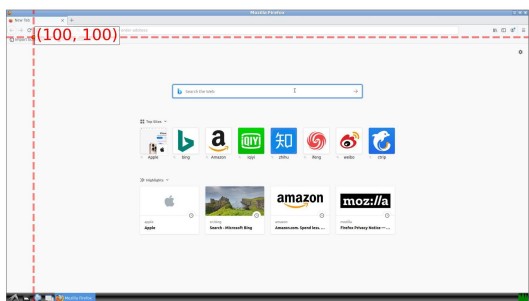

**Intent:** "Move the cursor to the top-left corner of the screen to begin navigation."

**Action:** "Move(100, 100)"

Figure 8: The *Qwen2.5-VL-3B* model demonstrates spatial awareness capabilities. When the intent specifies "top-left corner of the screen", the model outputs coordinates (100, 100), which accurately corresponds to the screen's top-left position.

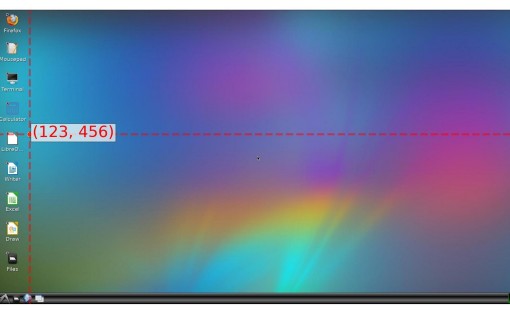

Intent: "To start the process of accessing something specific on this presumably not fully explored environment."

Action: "DoubleClick(123, 456)"

Figure 9: The *Qwen2.5-VL-3B* model followed the instructions in the prompt, attempting to utilize double click to interact with desktop icons. However, the specific coordinates were directly copied from the prompt text, indicating that, the language processing capabilities dominated over the visual localization abilities.

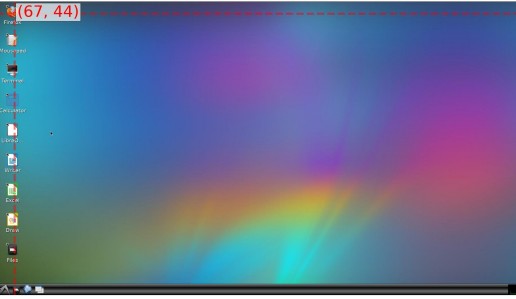

Intent: "To open an application, specifically Firefox for web browsing."

Action: "DoubleClick(67, 44)"

Figure 10: The original *Qwen2.5-VL-3B* model demonstrates the capability to correctly launch applications in a limited number of cases, with the intent descriptions aligning with the target actions. RL training can facilitate the emergence of such effective interactions.

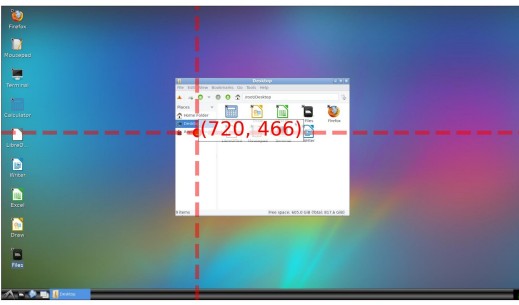

Intent: "I chose to open the 'Applications' folder because it may contain additional software and utilities that aren't immediately visible on the desktop, allowing me to explore more functionalities available in this environment."

Action: "DoubleClick(720, 466)"

Figure 11: The original *Qwen2.5-VL-72B* model, due to its larger model scale, demonstrates enhanced capability in interpreting action intentions, recognizing UI elements within images and generates more precise coordinate outputs.

# G  ABLATION ON EXPLORATION REWARD COMPONENTS

Our Exploration Reward designed comprises three categorical components: Instantaneous Change Reward, Subsequent Change Reward, and World Model Curiosity Reward. We examine how these three exploration reward components influence the training of exploration capabilities. We conducted ablation studies by removing each of these rewards individually, denoted as **w/o Instant**, **w/o Sequence**, and **w/o World Model**. Additionally, we included a ablation setting using only world model reward, labeled as **Only World Model**. Figure 12 presents a comparison of exploration metrics between these four ablation settings and *ScreenExplorer-3B*. The results demonstrate that the ablation setting **w/o World Model** exhibits the poorest performance in exploration metrics, with this group experiencing stagnation in the early exploration phase and showing limited potential for future exploration growth. Similarly, the group using **Only World Model** also showed suboptimal performance.

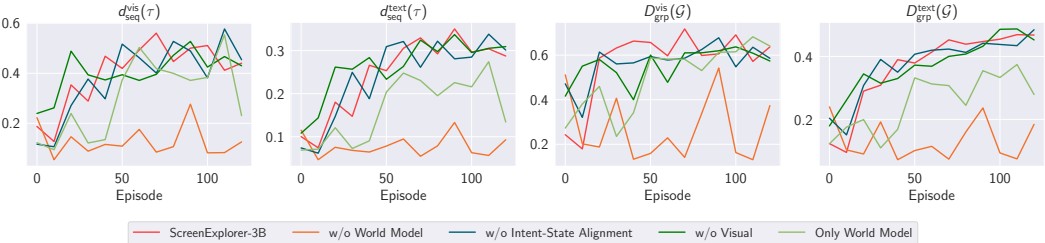

Figure 12: Impact of ablation studies on exploration metrics across different exploration reward components.

Comparing ablation settings **w/o Instant** and **w/o Sequence**, **w/o Instant** exhibited inferior performance in exploration metrics. Through analysis of training trajectories, we discovered that in the absence of immediate rewards, the model frequently became trapped in ineffective attempts, as illustrated by a typical case in Figure 13.

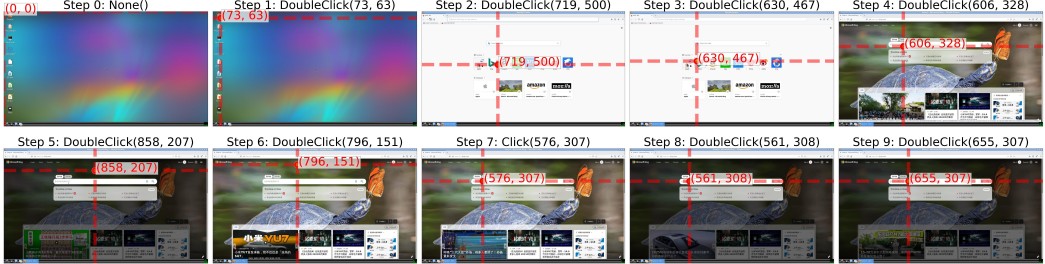

Figure 13: In a case study from the ablation group **w/o Instant**, while the agent successfully learned to open the browser, it demonstrated limited capability for deeper exploration.

We conducted two additional interesting ablation studies:

In the ablation setting **w/o Visual**, we removed all visual reward signals, specifically eliminating the $r_{inst}^{vis}$, $r_{seq}^{vis}$, and $r_{world}^{vis}$ reward components. This means that the model's exploration was driven solely by changes in on-screen text. Interestingly, even without rewards from visual information and relying exclusively on text-based reward signals, the agent was still capable of effective exploration in the GUI environment. We attribute this to the fact that the GUI is a complex environment containing abundant textual information.

Another notable finding is that when Intent-State Alignment rewards were incorporated during training, the model exhibited a stronger tendency to reference existing on-screen text in its intents, thereby enhancing the correlation between intent descriptions and screen content. Figure 14 presents a comparative analysis of intent descriptions generated by *ScreenExplorer-3B* and the model in **w/o Intent-State Alignment** ablation setting, where the latter excludes both $r_{des}$ and $r_{inter}$ reward components.

**w/o Intent-State Alignment**

Highlight key features and information about the product by clicking where the main feature is displayed

Move the cursor to the center of the screen and click the text area to explore the content further.

Clicking on one of the bookmarked websites could lead to further exploration or discovering new content related to the user's interests.

Move closer to product details by scrolling down or checking other sections like video or specifications

Move the mouse to another part of the screen to explore other elements and see if there is anything else useful or interesting to interact with.

Move towards understanding the product details and other options available for purchase

**ScreenExplorer-3B**

Click on the article titled "明晚北京又要下冰雹了明天北京再迎强对流天气" to explore and gather more information about the weather situation.

Clicking on the 'Hair Dryer' product link to explore options under the 'Top Categories in Kitchen Appliances' section.

Click on the first product titled '2.4GHz Wireless Gaming Headset' to explore further

Scroll down the page to explore more jeans options available

Click on the Amazon Essentials Skinny Jeans to explore further details and consider adding them to the cart

Click on the first result titled "侮辱雷军、小米、一博主被判道歉并赔6万元" by Tencent News.

Figure 14: Intent descriptions generated by *ScreenExplorer-3B* and ablation **w/o Intent-State Alignment** group. the *ScreenExplorer-3B* model demonstrates a stronger propensity to reference existing on-screen text when generating intents, resulting in more specific and contextually grounded intentions. In contrast, the **w/o Intent-State Alignment** group generates more generalized intents with less specific task objectives and minimal reference to on-screen textual content.

The ablation studies indicate that among all exploration rewards, the World Model reward is the most crucial, followed by the Instantaneous Change Reward. Furthermore, the incorporation of additional rewards can effectively modify the model's exploration preferences and Chain-of-Thought output patterns, providing valuable insights for future reward design and the construction of task-specific GUI datasets.

# H  CASE STUDY

In this section, we will present several cases during the training process of two size of *ScreenExplorer* agents to understand the development of exploration capabilities throughout the reinforcement learning training process.

## H.1  CASE STUDY OF SCREENEXPLORER-7B

Figures 15 to 19 present several case studies of the 7B model, demonstrating its superior GUI knowledge, environmental interaction and scene comprehension capabilities compared to the 3B model. Through our training process, we were able to enhance its exploratory behavior, enabling it to discover more diverse and deeper environmental states.

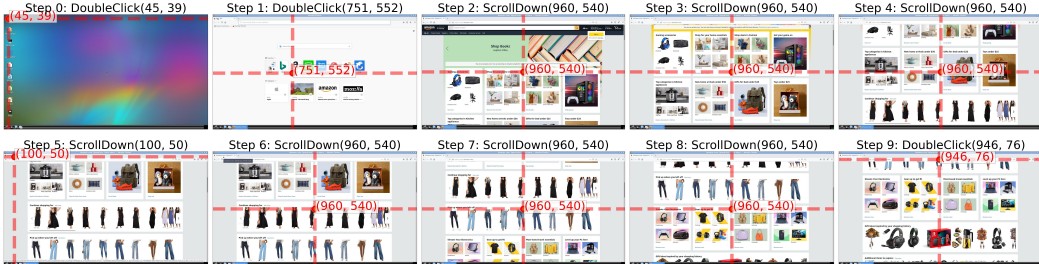

Figure 15: Episode-50 of *ScreenExplorer-7B*: Utilizing the ScrollDown command for web page navigation.

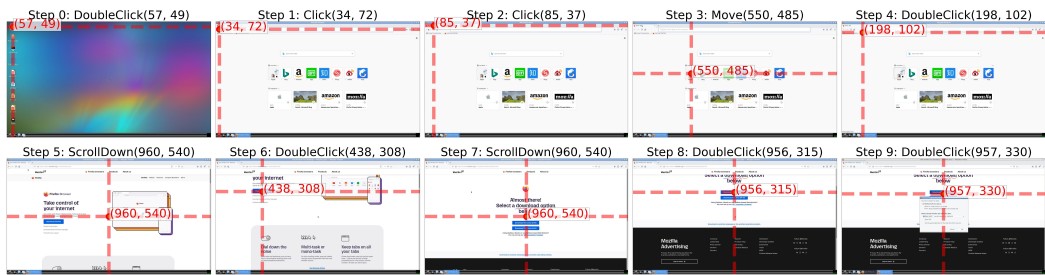

Figure 16: Episode-50 of *ScreenExplorer-7B*: The 7B model successfully completed a web browsing and software download sequence.

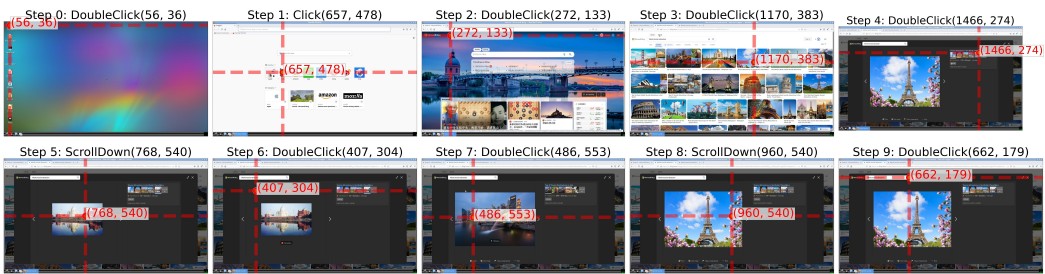

Figure 17: Episode-90 of *ScreenExplorer-7B*: The 7B model discovered that browsing through images yields higher exploration rewards.

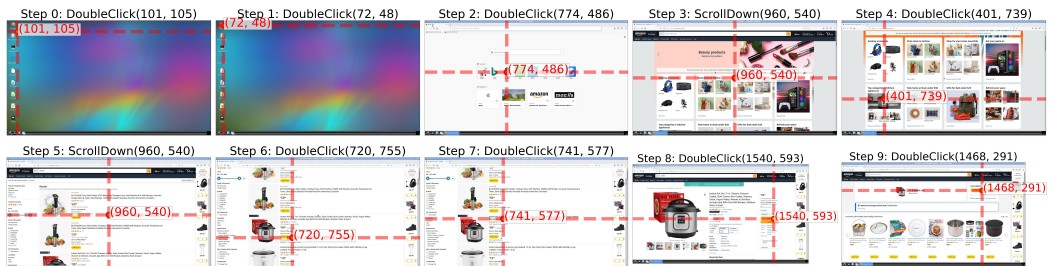

Figure 18: Episode-100 of *ScreenExplorer-7B*: The 7B model successfully completed a product purchase flow, from product selection to adding items to the shopping cart.

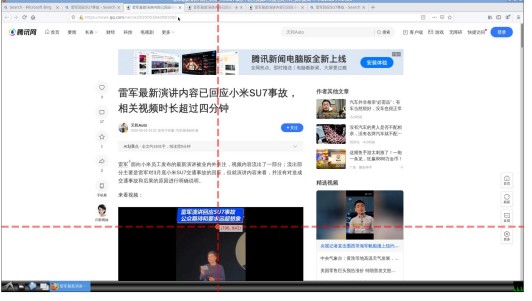

**Intent:** "To continue exploring the webpage and find relevant information about the incident and the corresponding video. Clicking the video link will allow us to watch the related content."
**Action:** "DoubleClick(796, 841)"

Figure 19: At episode 160, the 7B model demonstrated superior image reasoning capabilities and more precise action execution. In its intent description, it first comprehended that the news content pertained to an incident, then identified the potential presence of a "corresponding video" below, subsequently articulated a clear action intent of "clicking the video," and accurately targeted the video's location in its action output.

## H.2 CASE STUDY OF SCREENEXPLORER-3B

Figure 20 demonstrates the results of the *Qwen2.5-VL-3B* model interacting directly with the environment without RL training, where the model struggles to engage in effective interactions with the environment. Figure 21 to 28 present several case studies from *ScreenExplorer-3B* checkpoints.

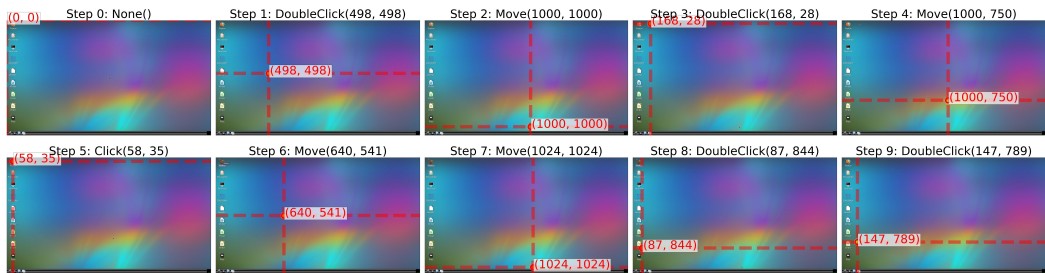

Figure 20: The original *Qwen2.5-VL-3B* model: The initial Qwen2.5-VL-3B model generated near-random coordinates, making it difficult to accurately click on icons and effectively interact with the environment, resulting in extremely low exploration rewards and diversity metrics.

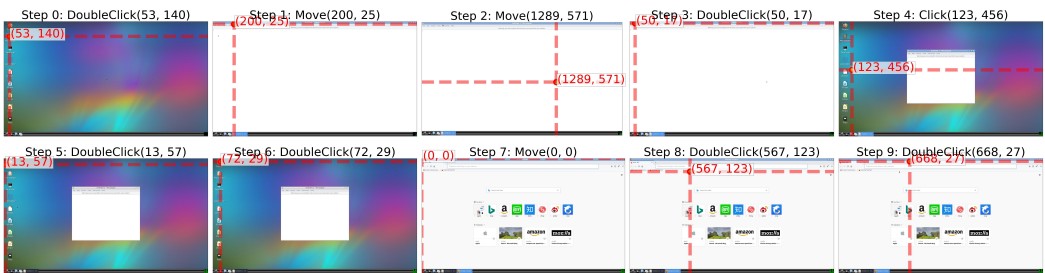

Figure 21: Episode-50 of *ScreenExplorer-3B*: The model demonstrated the capability to output commands for launching desktop applications, resulting in significant screen state changes, though it had not yet developed exploration behaviors in specific apps.

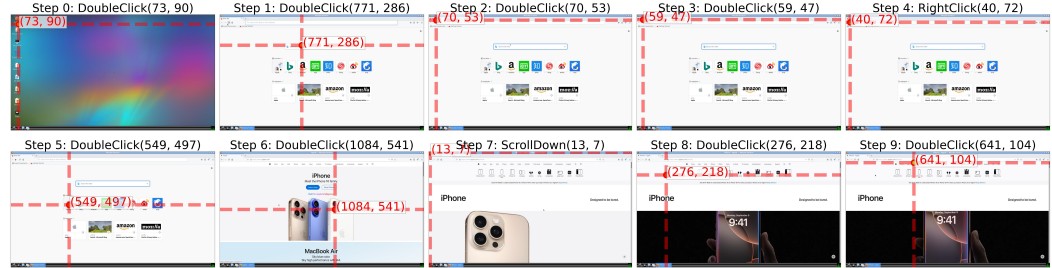

Figure 22: Episode-80 of *ScreenExplorer-3B*: The model demonstrated the ability to navigate from Firefox's homepage to explore a specific webpage.

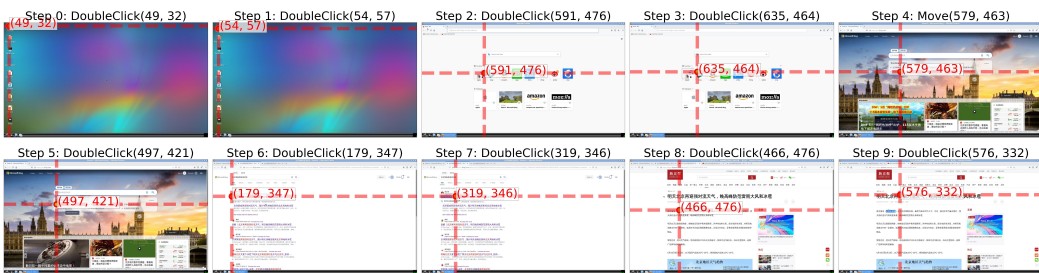

Figure 23: Episode-100 of *ScreenExplorer-3B*: The model demonstrated the capability to navigate through news feeds and access articles for reading.

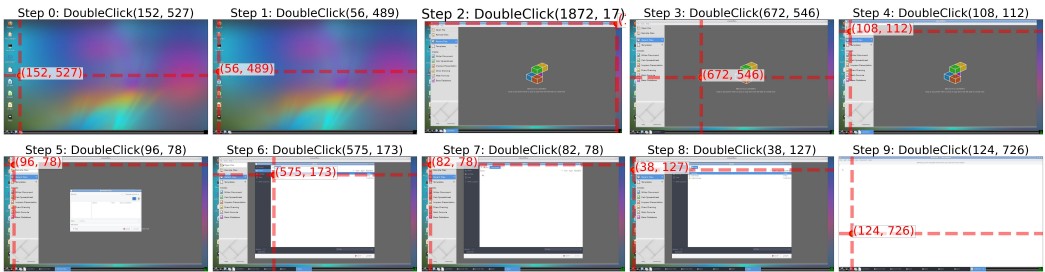

Figure 24: Episode-100 of *ScreenExplorer-3B*: The model was exploring the LibreOffice software interface.

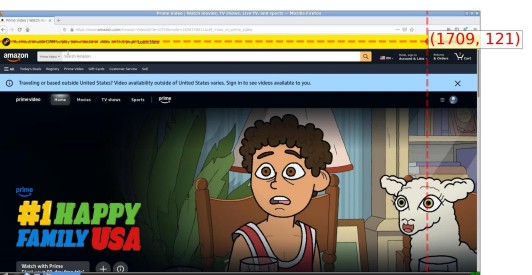

**Intent:** "First, I need to enable DRM to watch the stories about the children, which might be a prerequisite before exploring content further."

**Action:** "DoubleClick(1709, 121)"

Figure 25: At episode 70, the agent detected a DRM warning in the browser and observed that "stories about the children" were available for watching. Consequently, it opted to dismiss the DRM warning, demonstrating its ability to reason based on the screen state and make contextually appropriate decisions.

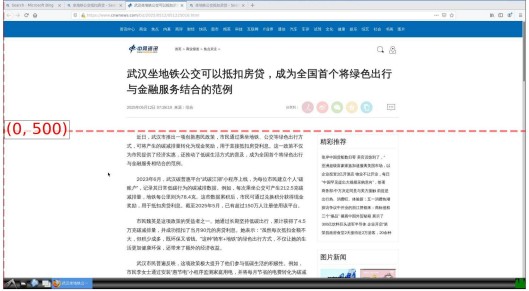

**Intent:** "Scroll down to read more content on the webpage"

**Action:** "ScrollDown(0, 500)"

Figure 26: At episode 80, the agent scrolled down on the news page to acquire additional information, demonstrating its capacity for content exploration.

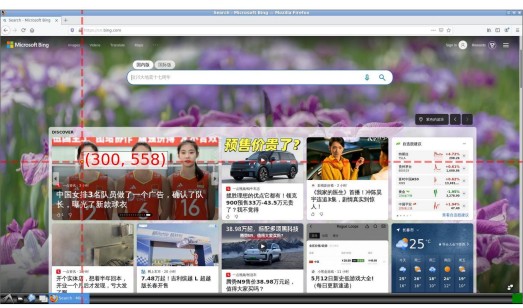

**Intent:** "Select the first news item about the Chinese women's volleyball team to explore more details"
**Action:** "DoubleClick(300, 558)"

Figure 27: At episode 100, the agent demonstrated cross-modal translation capabilities, recognizing Chinese news headlines from the visual input and generating corresponding English outputs in its intent description.

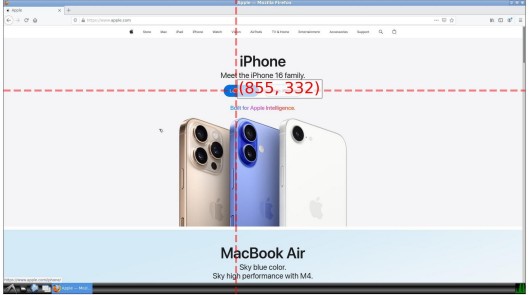

**Intent:** "Having reached the iPhone page at apple.com, the next logical step would be to learn more about the new iPhone models."
**Action:** "DoubleClick(855, 332)"

Figure 28: At episode 110, the intent description from *ScreenExplorer-3B* demonstrates its ability to recognize that the current page content pertains to "apple.com". Furthermore, the model specified the next operational plan in its intent, and subsequently executed actions targeting the correct button.

