# OpenReview forum: "Training a Vision-Language Model for Diverse Exploration in Open GUI World"
_ICLR.cc/2026/Conference — Submitted to ICLR 2026_

### Official Review · Reviewer_u3BX · 2025-10-30

**Soundness:** 3
**Presentation:** 3
**Contribution:** 3
**Rating:** 4
**Confidence:** 4

**Summary:**

This study proposes a reinforcement learning agent that autonomously explores GUI environments using a vision-language model. The proposed agent employs a reward function composed of nine components, including immediate, subsequent, and alignment rewards, together with a World Model–based curiosity reward to effectively explore unseen interfaces. To achieve more stable learning under the sparse and high-variance nature of GUI environments, the authors adopt Group Relative Policy Optimization (GRPO), demonstrating that it enables increasingly diverse exploration as training progresses.

**Strengths:**

- The study appropriately defines the observation and action spaces, which are essential aspects when applying reinforcement learning to web environments, and provides detailed explanations along with a plausible experimental setup.
- It conducts extensive experiments, including ablations, to analyze how visual reward signals and the world model influence exploration.
Defining a reward function in GUI environments is inherently challenging, but this work provides more combinations of the rewards than just simple single-step return rewards by introducing additional components such as the subsequent change reward, and empirically examines their effects on exploration through ablation studies.
- Given the sparse and highly fluctuating rewards characteristic of GUI environments, the choice to use GRPO instead of PPO to reduce variance is reasonable.

**Weaknesses:**

- This work uses visual/text diversity at the trajectory and group levels as its primary metrics, but it remains unclear whether these metrics truly correspond to meaningful web exploration rather than simply capturing random or superficial behavioral differences.
For instance, in Figure 5, the agent consistently selects the web browser across all episodes, and this could be due to the browser’s inherently higher visual diversity rather than intentional exploration, suggesting that the agent may have converged to a local optimum without exploring other diverse applications.

- The authors does not provide quantitative evaluations of the World Model’s prediction accuracy or qualitative visualizations comparing predicted versus actual screens.
Given that fine-grained predictive understanding is critical in such settings, presenting only the finding that the World Model helps during the cold-start phase, without evidence of its predictive fidelity, feels somewhat incomplete.

- Since several proposed reward components are directly tied to the evaluation metrics themselves (e.g., World Model–based reward), there is a risk that the agent’s exploration behavior becomes overly dependent on these self-referential metrics, potentially leading to biased or misleading exploration if the World Model is inaccurate.

**Questions:**

- Is there any way to verify the quantitative results of the World Model?
- In the GUI environment, does a sufficiently converged agent always choose the web browser?
If so, would this behavior correspond to a local optimum?

---

> ### Author Response · Authors · 2025-11-22
>
> ## 1. On whether visual/text diversity truly corresponds to “meaningful exploration”
>
> We appreciate the reviewer’s concern that visual and textual diversity metrics may not directly correspond to meaningful exploration. We chose visual and textual diversity as our primary metrics mainly for two reasons:
>
> - In terms of **availability and cross-environment generality**, screen images and OCR text are the most easily collectible representations in current GUI environments and are present across almost all systems and applications.
> - Many more structured representations (e.g., A11Y trees, HTML DOM, widget trees) are ultimately encoded as vectors or sequences when used to compute differences, which essentially reduces to measuring diversity in a **visual/textual feature space** as well.
>
> Regarding the phenomenon in Figure 5 that the agent increasingly prefers the browser, our observations are:
>
> - In the early stage of training, the model indeed moves among multiple applications such as terminal, file manager, and Office.
> - As training progresses, the browser, which naturally exhibits higher visual and textual variability, gradually becomes a more “rewarding” target under diversity-related rewards, so the agent tends to maintain longer interactions within the browser.
>
> We also want to add that, from the perspective of **“whether a particular downstream benchmark immediately benefits”**, certain states that appear “visually novel” may indeed not directly translate into performance gains on any specific benchmark task. However, this judgment is inherently **post hoc and task-dependent**: it is made only after we fix a particular set of downstream tasks. During training, ScreenExplorer does not know which GUI tasks it will be evaluated on in the future, and can only expand its reachable state space and repertoire of interaction patterns based on the intrinsic rewards provided.
>
> Therefore, we do **not** claim that this paper has fully solved the problem of “how to define and measure meaningful exploration.” Instead, our focus is to first **build a practically runnable, open-ended exploration framework** in real GUI environments without predefined tasks or manual external rewards, and to **systematically increase interface coverage, multi-step interaction structure, and behavioral diversity** under this setting. We believe that, on top of this broad-coverage exploratory corpus, adding task signals, automatic subgoal discovery, and transfer learning mechanisms is the right path toward answering the harder question of “which exploratory behaviors are most valuable for specific downstream tasks.”
>
> ## 2. On quantitative evaluation of the World Model and “predicted vs. actual screens”
>
> In our framework, the World Model plays a crucial role in improving exploration efficiency during the cold-start phase. We will include quantitative evaluation curves of the World Model, as well as representative “predicted vs. actual screen” visualizations in the appendix to more directly illustrate its effect.
>
> In the revision, we will add **quantitative curves of the World Model’s prediction metrics** as well as representative “predicted vs. actual screen” visualizations in the appendix to make its behavior more transparent.
>
> In addition, we would like to clarify the intended role of the World Model in our framework. In our design, the World Model is used **primarily as a curiosity signal**, in a spirit similar to Random Network Distillation (RND), rather than as a high-fidelity generative model. Its prediction error is leveraged to indicate which states are currently “hard to predict” and thus worth exploring, which is why it is especially effective in the cold-start phase.
>
> ## 3. On potential self-referential risk between rewards and evaluation metrics
>
> The reviewer is concerned that using a World-Model–based reward might create a self-referential loop if the same quantity is also used as an evaluation metric. We consciously avoid this issue by separating training and evaluation signals:
>
> - During **training**, the World Model’s prediction error is used purely as an **intrinsic curiosity reward**, encouraging the agent to visit states that are currently hard to predict.
> - During **evaluation**, we do **not** use the World Model’s prediction error as a metric. Instead, we rely solely on **external metrics that are independent of the World Model**, such as image-level and text-level diversity, to measure exploration diversity.
>
> Under this design, the sources of training signals and evaluation metrics are decoupled, so the same quantity is never used both as a reward and as a test metric. Even if the World Model is imperfect in some regions, it affects **which states are explored**, but does not bias our **evaluation** of exploration diversity, which is computed from independent visual/textual features.

---

> > ### Comment · Reviewer_u3BX · 2025-11-26
> >
> > I appreciate the authors' response.
> > However, I still have remaining concerns whether these metrics truly correspond to meaningful web exploration, with the quality of world model prediction.
> > Thus, I am keeping the original score.

---

### Official Review · Reviewer_WPCF · 2025-10-31

**Soundness:** 2
**Presentation:** 3
**Contribution:** 2
**Rating:** 4
**Confidence:** 4

**Summary:**

This paper presents ScreenExplorer, a vision-language agent designed for exploration and interaction within real, dynamic, and open-ended GUI environments. The framework integrates a curiosity-driven reward mechanism that leverages a learned world model to enhance exploratory behavior; a reinforcement learning pipeline based on RL; and an experience stream distillation procedure that improves adaptation and reduces reliance on manually curated datasets.

**Strengths:**

1.	The combination of RL, a learned world model, and experience stream distillation forms a coherent, reproducible pipeline for self-supervised GUI exploration.
2.	The world model, trained on paired image-text state transitions, introduces an intrinsic curiosity reward that improves cold-start exploration and advantage variance.
3.	The paper analyzes reward components, showing that removing the world model or alignment rewards degrades performance.

**Weaknesses:**

1.	Although the paper claims that the agent is “rewarded for both successful interaction and exploration novelty,” the implementation shows that “successful interaction” merely refers to producing syntactically valid JSON actions that alter the GUI state. There is no external or task-based success criterion (e.g., reaching a goal, executing a correct function, or completing a workflow). As a result, the learned policy optimizes purely intrinsic objectives without assurance that these behaviors translate into useful or goal-directed interactions. This limits the practical significance of the reported improvements in exploration diversity and weakens the claim that ScreenExplorer enhances “interaction capability.”

2.	The system is designed almost entirely around exploration incentives. Curiosity-based rewards and diversity metrics are maximized without complementary exploitation or goal conditioning. Consequently, the policy may overfit to visually novel yet semantically meaningless actions (a form of the “noisy-TV” problem acknowledged by the authors). This imbalance raises doubts about whether the learned behaviors can generalize to structured GUI tasks requiring planning or consistency. The paper would benefit from experiments showing how ScreenExplorer behaves when explicit goals or extrinsic feedback are introduced.

3.	Despite the “open-world” framing, all reported experiments appear confined to a limited set of GUI applications with similar layouts and interaction patterns. There are no results demonstrating generalization to unseen interfaces, visual themes, or operating-system contexts. Without evaluation on held-out GUI types, the claim of “generalizable exploration” remains speculative. Additionally, the paper does not discuss potential domain adaptation techniques or cross-environment fine-tuning strategies.

4.	The paper defines nine reward components spanning format validity, visual and textual novelty, intent alignment, and world-model curiosity. However, their relative weighting, normalization, and mutual interference are underexplored. The only detailed ablation concerns the world-model term; other components (e.g., intent alignment or diversity scores) lack sensitivity analyses. Without systematic tuning or normalization (beyond GRPO’s implicit standardization), there is a risk of reward hacking, where the model exploits specific reward structures instead of learning genuinely diverse behaviors.

**Questions:**

1.	Could the authors clarify whether any extrinsic (task-success) reward or evaluator signal exists, and how the agent’s utility is measured beyond diversity?
2.	How sensitive is performance to the weighting of the nine reward components? Have normalization or coefficient ablations been tested?
3.	What prevents the curiosity reward from degenerating into the “noisy-TV” effect (repeating visually novel but meaningless actions)?

---

> ### Author Response · Authors · 2025-11-22
>
> ## 1. On “successful interaction” and the lack of extrinsic task signals
>
> We agree with the reviewer that, in the current implementation, “successful interaction” is primarily defined as issuing syntactically valid JSON actions that lead to observable GUI state changes, rather than completing an explicit, externally defined task. This is closely related to the problem setting we deliberately adopt: in this work we focus on building a self-driven exploration system in open GUI environments that does not rely on manually designed tasks, rather than directly optimizing the success rate of a specific workflow.
>
> Even under this definition, we believe that “successful interaction” is a meaningful intermediate target rather than mere syntactic correctness:
>
> - First, learning **a stable distribution of syntactically valid actions that consistently change GUI states** from scratch is a necessary precondition for later adding task constraints and extrinsic rewards. If the agent cannot reliably issue valid actions that affect the environment, introducing more complex tasks and rewards would only exacerbate sparsity and instability.
> - Second, during training we observe a clear behavioral progression: the agent moves from “almost random clicks that rarely change the screen” to reliably **opening applications, browsing content, and navigating across interfaces**. These behavior patterns, as shown in our visualizations and statistics, increase the coverage of reachable states and produce longer, more coherent interaction chains, rather than being limited to syntactically valid but trivial JSON outputs.
> - Third, we introduce an **Intent–State Alignment reward** during exploration: the model generates an explicit `intent` field in its CoT, which can be viewed as a description of the intended subgoal at the current step. When the subsequent action and resulting state change are consistent with this intent, the agent receives additional reward. This provides a **scalable, weak form of task-style labeling** at each step without manual annotation, and enriches “successful interaction” to include not only format correctness and state change, but also **consistency between stated intent and actual behavior**.
>
> ## 2. On the ablations of reward component weights
>
> In the current submission, we already report several key ablation experiments in Appendix G, including **“w/o World Model,” “w/o Intent–State Alignment,” “w/o Visual,”** and **“Only World Model.”** These ablations help clarify the roles of different reward components in our framework without requiring an exhaustive sweep over all weight combinations.
>
> Concretely:
>
> - **World-Model reward and its importance in the cold-start phase.**
> Comparing “w/o World Model” with the full reward setting, we observe that removing the world-model curiosity term in the early (cold-start) stage significantly reduces advantage variance and slows down the expansion of exploration behavior, whereas including it clearly accelerates **the first-time coverage of new interfaces and the divergence between trajectories**. This is consistent with our analysis around Figure 4, and suggests that the world-model reward mainly acts as a “bootstrap” signal that improves early exploration efficiency, rather than just adding another numerical term.
>
> - **Intent–State Alignment and its effect on intent–action consistency.**
> In Figure 14, we compare the model’s `intent` field in the CoT with and without the Intent–State Alignment reward. With this reward, the generated intents become **clearer and more aligned** with the actual state transitions caused by subsequent actions, resulting in a higher proportion of samples where the intent and the semantic change in the GUI are consistent. We therefore view this component as primarily improving **intent–action consistency and the interpretability of interaction semantics**, rather than as a noisy extra diversity term.
>
> - **“Only World Model” vs. full reward: the contribution of visual/text change rewards to diversity.**
> In the “Only World Model” configuration, the agent can still reach some uncertain states driven by the world-model curiosity, but the **trajectory-level visual/textual diversity and cross-application coverage** are noticeably lower than under the full reward configuration. This indicates that visual- and text-change–based rewards provide **additional driving force for diversity and coverage** beyond what the world-model curiosity alone can offer.
>
> Overall, we agree with the reviewer that a fully systematic study of the weights, normalization, and interactions of all nine reward components is a valuable direction, but it is also a substantial undertaking. We believe, however, that the existing comparisons (**w/o World Model, w/o Intent–State Alignment, Only World Model vs. full reward**) already provide sufficient evidence to support our main claims about the functional roles of the key reward components in ScreenExplorer.

---

> > ### Author Response · Authors · 2025-11-22
> >
> > ## 3. On exploration-heavy incentives, “visually novel but semantically meaningless” Noisy-TV behavior, and reward hacking risks
> >
> > We fully share the reviewer’s concern that curiosity-driven exploration may lead to classic Noisy-TV–style behavior. In our experiments, we also observe related patterns:
> >
> > - In the early stage, the agent’s actions are close to random clicks and rarely induce any state changes.
> > - As training progresses, the agent gradually learns more structured exploratory behaviors, such as opening a browser and navigating across multiple pages.
> > - In the later stage, without external goal constraints, the agent can indeed develop a preference for a small subset of pages that can sustain high visual diversity over long horizons, which creates a form of reward path dependence and reduces diversity at the application level.
> >
> > It is important to emphasize that whether a state is “visually novel but not directly useful for a downstream task” is largely a **post-hoc judgment from the perspective of a particular task set**. During training, the agent does not know which specific tasks will matter in the future; it can only expand its reachable state space under the given intrinsic rewards. From this perspective, our goal in this paper is **not** to claim that we have fully solved the problem of “meaningful exploration,” but rather:
> >
> > - To propose and implement a **full pipeline that enables a VLM to conduct sustained self-driven exploration in real, open GUI environments**, without predefined tasks or manually specified external rewards, making “from-scratch large-scale GUI exploration” practically feasible.
> > - To automatically accumulate **large-scale, long-horizon interaction streams** during this process. Even though these streams are not yet fully mapped to specific tasks, they already have high potential value in terms of state-space coverage, interface structure, and multi-step interaction patterns.
> >
> > We view the role of ScreenExplorer’s policy model as analogous to that of **DeepSeek-R1-Zero**:
> >
> > - DeepSeek-R1-Zero is mainly responsible for “from-scratch exploration and process trajectory generation,” and its primary value is to provide process data for subsequent models (such as DeepSeek-R1), rather than achieving optimal performance on all tasks itself.
> > - Similarly, even if ScreenExplorer exhibits some Noisy-TV tendencies in later training, during the full training process from scratch it still **covers substantially more interfaces, longer interaction chains, and more diverse operation patterns than static VLM baselines**, providing unprecedented real GUI interaction data for subsequent distillation, behavior pattern discovery, and subgoal mining.
> >
> > From a longer-term perspective, we see this direction as a **path towards data scaling for GUI agents**:
> >
> > - On one hand, existing GUI benchmarks heavily rely on manually designed tasks and annotations, which makes it difficult to further increase task variety and data scale.
> > - On the other hand, by allowing a VLM to continuously self‑explore and self‑evolve in open environments, we can keep accumulating “native interaction streams” that resemble real user behavior, and then gradually turn them into supervised or RL training signals via post-hoc mining (e.g., automatic subgoal discovery, skill abstraction, and high-value trajectory selection).
> >
> > Therefore, we regard the Noisy-TV phenomenon as a **structural challenge intrinsic to this setting**, rather than evidence that the proposed pipeline is flawed. Our current work provides:
> >
> > - A reproducible end-to-end framework that connects exploration rewards, world-model-based curiosity, and experience-stream distillation in real GUI environments, and
> > - Concrete empirical observations of Noisy-TV–like behavior in large-scale open-world self-exploration,
> >
> > which we hope will serve as a solid basis for future work that introduces stronger task signals, post-hoc trajectory filtering and reweighting, and more robust curiosity objectives to further narrow the gap between “visual novelty” and “task relevance.”

---

### Official Review · Reviewer_YR9L · 2025-11-01

**Soundness:** 1
**Presentation:** 3
**Contribution:** 2
**Rating:** 2
**Confidence:** 4

**Summary:**

ScreenExplorer introduces a VLM-based agent trained via reinforcement learning in real GUI environments to enable autonomous exploration without relying on predefined task structures. The key innovation is a hierarchical reward system combining state-change rewards, world model-based curiosity signals, and intent-state alignment rewards for GRPO. Experiments demonstrate improved exploration diversity compared to static baselines, with the model improving from worst to best performer through RL training.

**Strengths:**

- Addresses the important but under-explored challenge of autonomous GUI exploration in open-ended environments, moving beyond task-specific training.
- The multi-faceted reward function elegantly combines immediate feedback (format, instant change), long-term diversity (subsequent change), curiosity (world model predictions), and grounding (intent-state alignment).
- Clear improvements in exploration diversity metrics, with ScreenExplorer-7B achieving 0.55 average diversity compared to 0.25 for GPT-4o and 0.43 for Qwen2.5-VL-72B.
- The world model curiosity reward successfully addresses the exploration cold-start problem by increasing advantage variance (Figure 4), enabling the 3B model to overcome initial learning barriers.

**Weaknesses:**

- The paper critically lacks evaluation on actual GUI tasks. While exploration diversity is measured extensively, there's no evidence that this exploration improves performance on established benchmarks like WebArena, VisualWebArena, or Mind2Web. This is a fundamental gap. Exploration is only valuable if it improves task performance.
- The diversity metrics (visual/textual sequence diversity) don't clearly correlate with useful exploration. The agent might be exploring irrelevant states (e.g., clicking random news articles) without learning transferable skills.
- Evaluation is restricted to a single Linux desktop environment. No evidence of generalization to other platforms (Windows, macOS, mobile) or complex web applications.
- While ablations show which rewards contribute to diversity, they don't demonstrate which exploration behaviors actually help downstream task learning. The "noisy TV problem" is mentioned but not thoroughly addressed.
- The filtering process for experience streams relies on GPT-4o-mini or manual curation, but there's no analysis of what makes exploration trajectories valuable for learning.

**Questions:**

- What is the performance on established benchmarks? How does ScreenExplorer perform on WebArena, VisualWebArena, or Mind2Web after exploration pre-training? Does exploration diversity correlate with task performance? Can you show that models with higher exploration diversity actually perform better on downstream GUI tasks?
- How does the approach handle task-specific fine-tuning? After exploration pre-training, how should the model be adapted for specific tasks?
- What prevents meaningless exploration? How do you ensure the agent explores task-relevant states rather than just clicking randomly to maximize state changes?
- How does exploration transfer across environments? Does exploration in Linux desktop environments transfer to web or mobile applications?
- What is the quality of distilled behaviors? Do distilled models learn meaningful exploration strategies or just memorize trajectories?

---

> ### Author Response · Authors · 2025-11-21
>
> ## 1. On downstream tasks and benchmark results
>
> We strongly agree with the reviewer that “exploration ultimately needs to translate into task performance.” In this work, we deliberately focus on **exploration and effective interaction capabilities in open environments**, rather than directly performing end-to-end optimization on specific benchmarks such as WebArena or VisualWebArena. The main reason is that, given current base model capabilities and extremely sparse rewards, our attempts at task reinforcement learning in environments like OSWorld could hardly roll out enough successful trajectories, and end-to-end training struggled to converge stably. This phenomenon itself motivated the setting of this paper: we first use multiple exploration rewards to let the model learn to reliably change GUI states in real environments, expand the reachable region, and accumulate reusable interaction patterns. On top of that, we then introduce task-oriented fine-tuning or hybrid rewards. In the revision, we plan to add several small-scale downstream task experiments to provide initial evidence for the feasibility of “exploration pre-training + task fine-tuning.”
>
> We believe that this paper already constitutes a fairly complete exploratory study in this direction, while a systematic characterization of “the relationship between exploration metrics and downstream task performance” would itself require a separate, long-term piece of work.
>
> ## 2. On how to connect to task‑specific fine‑tuning
>
> Our setting assumes that, in the absence of predefined task structures, the agent performs **self‑exploring / self‑evolving** in open GUI environments. The goal of this phase is to obtain exploration trajectories and experience streams that cover a broader and more structured state space, rather than directly optimizing the success rate of a particular task. The gap between “exploration” and “task completion” still requires further study.
>
> Our envisioned path for task fine-tuning is: during exploration, we annotate or automatically mine “subgoals / skills” from parts of the trajectories, then use a well-trained world model to evaluate the achievement of these subgoals, turning them into external rewards for reinforcement learning. Existing GUI benchmarks depend heavily on manually designed tasks and manually annotated evaluation metrics, making it difficult to scale up task variety and evaluation cost. In contrast, our setting aims to **simultaneously expand the space of learnable tasks and the corresponding evaluation signals** by automatically proposing candidate subgoals in open environments and having the world model evaluate them.
>
> ## 3. On “meaningful exploration” and the Noisy TV problem
>
> We fully agree with the reviewer’s concern about the noisy-TV problem. We also observed similar phenomena in our experiments: in the early stages of training, the model’s actions resemble random clicking and hardly induce any meaningful state changes. As training progresses, the agent gradually learns more structured exploration behaviors such as opening a browser and browsing web pages. Later, in the absence of external goal constraints, the model starts to prefer a subset of pages that can generate high visual diversity for a long time, which leads to a noisy‑TV–like reward path dependence and actually suppresses diversity at the application level.
>
> We view this phenomenon not as a “bug” in our method, but as a **structural challenge** that naturally emerges when curiosity-driven mechanisms are deployed in **large-scale, open GUI environments** under a self‑explore setting. In the Discussion section, we explicitly highlight the noisy‑TV phenomenon as a starting point for future work on “how to introduce more appropriate constraints and task signals into open-ended exploration.”
>
> In future work, we plan to mitigate this over-dependence on noisy-TV–like states through random initial states, larger-scale parallel environments, periodic resets, and lightweight coupling with external tasks or databases.
>
> ## 4. On cross-environment and cross-platform generalization
>
> To address the reviewer’s concern about cross-platform generalization, in the subsequent response and revision we will add an exploration experiment conducted in a Windows desktop environment. More large-scale analyses of transfer to web environments, multi-window systems, and mobile platforms will be systematically pursued in future extended work.

---

> > ### Author Response · Authors · 2025-11-21
> >
> > ## 5. On distillation and the quality of distilled behaviors
> >
> > Regarding the “quality of distilled behaviors,” our use of GPT‑4o‑mini and manual curation as filtering strategies is primarily aimed at selecting action data where the intent and action are well aligned. In our setting, the model’s predicted intent can be directly rewritten by GPT‑4o‑mini into the task goal for the current step, which effectively gives us action data with task-style annotations “for free” during exploration. In addition, compared with the online RL policy, the distilled policy preserves a comparable level of trajectory‑level visual and textual diversity.
> >
> > As for whether the distilled model “learns meaningful exploration strategies or just memorizes trajectories,” we agree that SFT alone is unlikely to fully “teach” a distilled model a complete exploration strategy. What it can reliably learn, however, are the effective interaction patterns and task‑completion behaviors that have been accumulated in the exploration trajectories. This allows the distilled model, at inference time, to replay representative exploration patterns and complete certain tasks at a lower cost, rather than starting the search process from scratch.

---

> > > ### Comment · Reviewer_YR9L · 2025-11-25
> > > **Thank you and follow up**
> > >
> > > I appreciate the author’s response and I do see the value of improving exploration as the authors are proposing.
> > >
> > > However, without a task performance experiment, it’s hard to know if the exploration behavior you are measuring will necessarily lead to better policy learning.
> > >
> > > Is there evidence you can point to that performing better on the exploration metrics reported (and embedding models you used to compute them) leads to better policy learning (especially in web domains)?
> > >
> > > Would it not be possible to measure pass@k on an established dataset to validate the exploration policy more often reaches successful states?

---

### Official Review · Reviewer_oiXs · 2025-11-02

**Soundness:** 3
**Presentation:** 3
**Contribution:** 3
**Rating:** 6
**Confidence:** 3

**Summary:**

This paper proposes ScreenExplorer, a vision-language model trained via reinforcement learning for open-world GUI exploration. The agent learns to interact with real desktop interfaces without predefined goals, driven by multi-term rewards that capture both state changes and semantic alignment. A world-model-based curiosity signal promotes novelty, and experience stream distillation helps consolidate diverse experiences for continual improvement. Experiments in a Linux GUI environment show consistent gains in exploration diversity and novelty compared to strong VLM baselines.

**Strengths:**

- The paper explores a fresh and timely problem (open-world GUI exploration), moving beyond fixed task datasets toward agents that can learn to explore software interfaces on their own through curiosity-driven interaction.
- It proposes a well-designed framework that combines a world-model-based curiosity signal, multi-part state-change rewards, and GRPO optimization, with experience stream distillation helping the agent gradually improve through self-collected experience, this is intuitive and effective.
- Experiments in a realistic Linux desktop environment show noticeable gains in exploration diversity and novelty compared to strong VLM baselines, and the analyses clearly demonstrate how the curiosity signal and alignment rewards contribute to the results.
- The paper is clearly written and well-organized.

**Weaknesses:**

- The evaluation is done only in a custom Linux GUI environment with a small set of apps and layouts. While this makes the study controlled and clean, it doesn’t reflect the variety and complexity of real-world interfaces like web or multi-window systems. It’s therefore unclear if the exploration policy and curiosity module would still work well in broader or more realistic GUI settings.

- The ablation studies focus on removing reward terms or curiosity signals but don’t test the effect of experience stream distillation, reward weights, or world-model design. As a result, it’s hard to tell which parts of the system actually drive the performance gains, rather than the whole pipeline working together.

-  While the framework is well-designed, many parts (like the curiosity module, GRPO training, and distillation) are adapted from existing methods. The paper mainly integrates these components rather than introducing new techniques, so the overall novelty feels more like a solid system combination than a new algorithmic idea.

**Questions:**

See weakness

---

> ### Author Response · Authors · 2025-11-21
>
> 1. Regarding the concern of “only in a custom Linux GUI environment,” we agree with the reviewer’s point about the lack of environmental diversity. In the subsequent response and revision, we will add an experiment conducted in a Windows desktop environment, showing the trend of exploration behavior across heterogeneous operating systems to provide initial evidence of the method’s applicability on different GUI platforms.
>
> 2. In Figure 12 of Appendix G, we have already presented ablation results for several reward components, including removing the world model, removing the intent‑state alignment reward, removing the visual component, and using only the world-model reward, in order to analyze how each component affects exploration behavior. We agree with the reviewer that systematically analyzing the sensitivity of experience stream distillation, reward weights, and world-model structure is a very valuable direction. Due to space limitations, in the current version we only include the most essential set of ablation experiments and provide the corresponding curves in the appendix. More fine-grained analyses will be further explored in extended or follow-up work.
>
> 3. Regarding the comment that “the method mainly systematizes existing components and has weak algorithmic novelty,” we partially agree with the reviewer’s assessment of the sources of our components, but would like to further clarify the contributions of this work: on the one hand, most existing GUI agent works rely on manually annotated tasks and trajectories, and are evaluated on relatively closed benchmark datasets. Our setting instead lets a VLM continuously and autonomously explore in a real desktop environment and accumulate experience streams, constructing a runnable self‑explore / self‑evolve framework without any predefined task structure, which is still largely unexplored in the GUI agent literature. On the other hand, around this setting we design and implement a complete pipeline, from exploration reward design and RL optimization to experience stream distillation and visualization analysis, and in actual runs we expose issues such as the “noisy TV” phenomenon that arise in large-scale open-world self-driven exploration. This provides a reproducible experimental foundation for future work that introduces task signals or designs more robust curiosity mechanisms under this setting. Therefore, we prefer to position this work as a systematic methodological and problem-setting exploration for real GUI environments, filling an engineering and system-level gap in enabling current methods to run stably in open environments, rather than as a simple stacking of existing components.

---

### Author Response · Authors · 2025-11-21
**Explaining Our Problem Setting and System-Level Focus Beyond Benchmark Task Performance**

We thank all reviewers for their careful evaluations and constructive feedback. We first clarify the **core goal and problem setting** of this work: ScreenExplorer is not primarily designed to “directly improve task success rates on existing GUI benchmarks.” Instead, it targets a long-neglected aspect in current GUI agent research: **how to enable large models to autonomously explore and interact stably in open environments.**

Most existing GUI works rely on manually designed tasks and annotated trajectories, and are evaluated on relatively closed benchmarks. In contrast, when humans face a new system or application, they typically begin with exploration to identify “reachable goals” and “reusable interaction structures,” and only then move on to solving specific tasks. We argue that, for large models to truly acquire the ability to “adapt to any new interface,” we first need a mechanism that allows them to **continuously self-explore and accumulate experience in real GUI environments**, rather than only performing supervised or RL fine-tuning on a fixed set of tasks.

In practice, we attempted task-level reinforcement learning directly in environments such as OSWorld. However, due to limited base model capability and extremely sparse rewards, it was almost impossible to roll out successful trajectories and obtain stable positive returns, so end-to-end optimization did not converge. This practical bottleneck directly motivated the setting in this paper: we instead design multiple exploration rewards and a world‑model‑based curiosity signal, and **prioritize the “precondition problem” of whether the agent can establish effective interaction with the environment and continuously discover new states**, laying the behavioral and data foundation for later introducing task-oriented rewards and constructing large-scale task spaces.

Therefore, we position this work as a **problem-setting and system-level exploration**: we propose and validate a feasible framework that enables a VLM to perform self-driven exploration in real GUI environments, systematically analyze how different exploration rewards and curiosity signals shape behavior patterns, and build a closed loop from long-horizon exploration to model updates via experience stream distillation. We believe this line of work is complementary to research that “directly improves task success rates on existing benchmarks”: the former provides the fertile ground for scalable adaptation of large models in open-world settings, while the latter can then build on this foundation by adding concrete tasks and evaluations.

---

### Meta-Review · Area_Chair_aFMa · 2026-01-01

**Summary:**

This submission introduces ScreenExplorer, a VLM-based GUI agent trained with reinforcement learning to perform open-ended exploration in real desktop environments without predefined tasks. The core idea is to use a multi-term intrinsic reward (state-change, diversity, intent–state alignment) plus a world-model–based curiosity signal to overcome cold-start, and to close the loop via experience stream distillation. Reviewers agree the problem setting is timely and the system is thoughtfully engineered, but the review record repeatedly emphasizes several decision-driving gaps: (i) the paper provides no downstream task/benchmark evidence that exploration improves practical GUI competence, (ii) it is unclear whether the proposed diversity/novelty metrics reflect meaningful exploration rather than superficial novelty or local optima (and the method may be vulnerable to noisy-TV/reward hacking), (iii) evidence for generalization is limited because evaluation is confined to a custom Linux environment, and (iv) key components (world model fidelity, reward weighting sensitivity, distillation contribution) are not fully isolated or validated via ablations/sensitivity analyses within the submission and discussion record. While the rebuttal clarifies the intended scope and motivates exploration-first due to sparse task rewards, it primarily describes planned additions rather than providing the missing evidence within the discussion thread, so the main uncertainties remain. Overall, while one reviewer remains marginally positive, the outstanding gaps cited above keep the paper below the bar.

**Reviewer Concerns:**

### Addressed by the rebuttal (partially)

- Problem setting clarification (exploration-first): Authors clearly explain why they focus on autonomous exploration rather than immediate task success (task RL not converging under sparse rewards), and position this as a prerequisite stage.

- Noisy-TV acknowledged: Authors discuss the phenomenon and propose mitigation directions (resets, parallel envs, random initial states, coupling with external tasks), but do not provide demonstrated mitigation results in the discussion record.

- Ablation references: Authors point to existing appendix ablations (e.g., removing world model, removing alignment reward) and explain intended roles of components.

- Separation of reward vs evaluation signals: Authors state evaluation metrics are computed independently of the world-model error used in reward.

### Still remaining (based strictly on the discussion record)

- Downstream task performance evidence: No concrete results on established task benchmarks or task-style evaluations are provided in the discussion; key reviewers treat this as fundamental.

- Metric validity / usefulness linkage: No demonstrated correlation between diversity metrics and improved task competence; concerns about superficial novelty/local optima remain.

- Generalization claims: Cross-platform/cross-environment evidence is not shown (Windows experiment is proposed, not reported).

- World model fidelity: Quantitative/qualitative prediction accuracy (“predicted vs actual”) is promised but not provided in the thread; concerns persist.

- Component attribution & sensitivity: No systematic analysis of reward weights, distillation effect magnitude, or world-model design choices beyond limited ablations.

**Reviewer Scores:**

- Reviewer oiXs (score 6): No explicit indication of a score change in the discussion; likely remains marginally positive. Rebuttal acknowledges limitations and proposes broader experiments but does not provide new evidence in-thread.

- Reviewer YR9L (score 2): No explicit indication of a score change in the discussion; reviewer follow-up reiterates that without task-performance experiments, usefulness is unclear.

- Reviewer WPCF (score 4): No explicit indication of a score change in the discussion; core concerns remain (intrinsic-only success definition, reward hacking/noisy-TV risk, generalization, weight sensitivity) without new in-thread evidence.

- Reviewer u3BX (score 4): No change (explicitly keeps score after rebuttal).

---

### Decision · Program_Chairs · 2026-01-26

Reject